# High-throughput RNA structure probing reveals critical folding events during early 60S ribosome assembly in yeast

Elena Burlacu[1], Fredrik Lackmann[2], Lisbeth-Carolina Aguilar[3], Sergey Belikov[2], Rob van Nues[4], Christian Trahan[3,5], Ralph D. Hector[6], Nicholas Dominelli-Whiteley[7], Scott L. Cockroft[7], Lars Wieslander[2], Marlene Oeffinger[3,5,8] & Sander Granneman [1]

While the protein composition of various yeast 60S ribosomal subunit assembly intermediates has been studied in detail, little is known about ribosomal RNA (rRNA) structural rearrangements that take place during early 60S assembly steps. Using a high-throughput RNA structure probing method, we provide nucleotide resolution insights into rRNA structural rearrangements during nucleolar 60S assembly. Our results suggest that many rRNA-folding steps, such as folding of 5.8S rRNA, occur at a very specific stage of assembly, and propose that downstream nuclear assembly events can only continue once 5.8S folding has been completed. Our maps of nucleotide flexibility enable making predictions about the establishment of protein–rRNA interactions, providing intriguing insights into the temporal order of protein–rRNA as well as long-range inter-domain rRNA interactions. These data argue that many distant domains in the rRNA can assemble simultaneously during early 60S assembly and underscore the enormous complexity of 60S synthesis.

[1] Centre for Synthetic and Systems Biology (SynthSys), University of Edinburgh, Edinburgh EH9 3BF, UK. [2] Department of Molecular Biosciences, The Wenner-Gren Institute, Stockholm University, SE-106 91 Stockholm, Sweden. [3] Institut de Recherches Cliniques de Montréal RNP Biochemistry laboratory, Systems Biology Axis, Montréal, QC H2W 1R7, Canada. [4] Institute of Cell Biology, University of Edinburgh, Edinburgh EH9 3FF, UK. [5] Université de Montréal Département de Biochimie et Médecine Moléculaire Faculté de Médecine, Montréal,, QC H3C 3J7, Canada. [6] Institute of Neuroscience and Psychology, University of Glasgow, Glasgow G12 8QB, UK. [7] EaStCHEM School of Chemistry, University of Edinburgh, Joseph Black Building, David Brewster Road, Edinburgh EH9 3FJ, UK. [8] McGill University Division of Experimental Medicine Department of medicine, Montréal, QC H4A 3J1, Canada. Fredrik Lackmann and Lisbeth-Carolina Aguilar contributed equally to this work. Correspondence and requests for materials should be addressed to S.G. (email: sgrannem@staffmail.ed.ac.uk)

Ribosome assembly in the yeast *Saccharomyces cerevisiae* is a complex and energy-consuming process. It starts with the transcription of recombinant DNA by RNA polymerase I and the co-transcriptional packaging of nascent pre-ribosomal RNA (rRNA) into a 90S complex[1, 2]. Cleavage of the nascent transcript at site $A_2$ (Fig. 1a) separates the large particle into precursors of the 40S and 60S subunits.

Assembly of the 60S subunit involves processing of two rRNAs (5.8S and 25S rRNAs), and the incorporation of the third rRNA (5S rRNA) as a separate ribonucleoprotein complex[3]. Following cleavage of the nascent transcript at $A_2$, the $27SA_2$ is converted into 27SB pre-rRNAs, either via cleavage at $A_3$ followed by exonucleolytic trimming to $B1_S$ or by cleavage at $B1_L$. The 27SB pre-rRNAs are cleaved at site $C_2$ in the second internal transcribed spacer (ITS2) region, generating 7S pre-rRNA and a 5′ extended 25S precursor (25.5S) (Fig. 1a). These are subsequently processed in the nucleus, and undergo final maturation steps after their export to the cytoplasm to form mature 5.8S and 25S rRNAs, respectively[4–6].

Pre-rRNA processing and folding requires the incorporation of many ribosomal proteins (r-proteins) and the activity of hundreds of assembly factors. Many of the r-proteins initially bind with low affinity, and their association becomes increasingly more stable as the assembly process proceeds[7]. This gradual assembly of r-proteins is regulated by ribosome assembly factors that chaperone their timely incorporation[7], and make the assembly process more efficient by preventing misfolding or kinetic traps that could lead to the formation of defective ribosomes[8].

Although the dynamics of the binding and dissociation of assembly factors has been studied in detail, still relatively little is known about the order of rRNA-folding steps, how assembly factors influence rRNA folding, and how this regulates the timely assembly of r-proteins. Recent biochemical and cryo electron microscopy (cryo-EM) studies have provided important insights into restructuring events in late pre-60S complexes[9–14], however, still little is known about rRNA folding during the nucleolar stages. To address this, we performed high-throughput RNA structure probing analyses (ChemModSeq[15]) on purified nucleolar pre-60S particles isolated from the yeast *S. cerevisiae*. These data enabled us to generate a step-by-step model of the changes in rRNA flexibility during the nucleolar stages of 60S assembly. Strikingly, our data indicate that the majority of the observed rRNA restructuring events take place during the conversion of $27SA_2$ to 27SB. During this stage, the 5.8S rRNA undergoes major structural rearrangements. These events coincide with the stable integration of r-proteins that are part of the polypeptide exit tunnel and the release of assembly factor Rrp5 that plays an important role in chaperoning rRNA folding[16]. Furthermore, we provide the first comprehensive structural analysis of ITS2 secondary structure, which demonstrate that ITS2 forms a highly compact structure, consistent with recent phylogenetic analyses[17]. Finally, comparing our nucleotide flexibility maps with the crystal structure of the 60S subunit enabled us to make predictions about the timing of the formation of r-protein-rRNA interactions as well as the formation of many long-range inter-domain interactions within the 60S subunit rRNAs.

Collectively, the results provide insights into the temporal order of rRNA restructuring events during 60S assembly and present a very useful resource for the design of more focussed functional analyses.

## Results

**Purification of pre-60S intermediates.** To measure RNA structural differences between large subunit (LSU) pre-rRNA species, we purified pre-60S particles using TAP-tagged bait proteins previously shown to be incorporated in complexes containing 35S (Mrd1), $27SA_2$ (Rrp5), and 27SB (Nsa2) rRNAs (Fig. 1a, b)[18–20]. After IgG affinity purification, co-precipitated pre-rRNAs were resolved on an agarose gel and the RNAs of interest were excised from the gel and purified. To assess the homogeneity of the purified pre-rRNA species, primer extension analysis was performed (Fig. 1b). The data show that Rrp5-TAP predominantly co-precipitated $27SA_2$ pre-rRNA (Fig. 1b, lane 1) and Nsa2-TAP almost exclusively co-precipitated 27SB pre-rRNA (Fig. 1b, lane 2). As TAP purifications with 90S-associated factors did not generate sufficient 35S pre-rRNA for high-throughput pre-rRNA structure probing analyses (data not shown), we used a TAP-tagged Mrd1 mutant strain (Mrd1Δ5) that is kinetically delayed in 35S pre-rRNA processing but does not noticeably affect 27S pre-rRNA levels (Supplementary Fig. 1a, b)[20, 21].

We also performed label-free quantitative mass spectrometry (MS) on the 60S pre-ribosomal particles purified using Rrp5 and Nsa2 as baits as well as 80S ribosomes, isolated using the translation initiation factor Fun12 (Fig. 1c–e, Supplementary Fig. 1c–e). We did not analyze the proteome of the Mrd1Δ5 particles as LSU assembly factors are likely highly under-represented in this complex, making the direct comparison with 60S particles complicated. The pre-60S MS data (Supplementary Data 1) are in excellent agreement with previous proteomic analyses of Rrp5- and Nsa2-associated complexes[14, 22–24]. Early 90S-associated assembly factors only co-precipitated with Rrp5, consistent with Rrp5 being part of both the 90S pre-ribosomes and early pre-60S intermediates[25], while Nsa2 associates with 27SB pre-rRNA containing pre-60S intermediates[26].

Conversion of $27SA_3$ to 27SB requires 12 proteins that are interdependent for their assembly, and which are referred to as $A_3$ factors (Fig. 1a)[27]. Cleavage of 27SB at site $C_2$ within ITS2 requires a set of assembly factors termed B-factors[28] (Fig. 1a). Interestingly, most $A_3$ factors (Nop7, Erb1, Ytm1, Rlp7, Cic1, and Nop15) were readily detectable in both Rrp5 and Nsa2 particles (Fig. 1c, d), suggesting that these proteins do not dissociate once 27SB has been generated. Notably, the $A_3$ factors were significantly enriched over the B factors in the Rrp5 pre-ribosomes (Fig. 1c), while in the Nsa2 particle, $A_3$ and B factors were present in comparable amounts (Fig. 1d). Assembly of the B factors takes place in two stages[28]; some B factors bind to the early 35S and $27SA_2$ pre-rRNA containing pre-ribosomes (Nop2, Nip7, Rpf2, Rrs1, Rlp24, Nog1, Tif6, Mak11, Spb4, and Dpb10), while others bind just prior to $C_2$ cleavage (Nsa2 and Nog2). Consistent with this, the early B factors were detected in both the Rrp5 and Nsa2 pre-60S intermediates, whereas the later B factors were predominantly present in the Nsa2 particles (Fig. 1c, d).

Further comparison between the r-protein content of the Fun12-, Rrp5-, and Nsa2-associated particles indicated that the purified pre-60S intermediates contained a similar r-protein composition, although not all r-proteins could be detected with high confidence due to a dynamic range detection limit of the instruments. Note that in this manuscript, we combine both the old and new nomenclature for the yeast r-proteins[29,30]. The r-proteins that are readily detected in very early 60S pre-ribosomes, such as L3/uL3 and L4/uL4 were identified in equivalent amounts in Rrp5 and Nsa2 pre-60S intermediates. Some r-proteins were slightly more abundant in the Nsa2 particle compared to the Rrp5 particle. This modest imbalance could be explained by the fact that r-proteins become more stably incorporated into pre-ribosomes as the maturation progresses[7, 31].

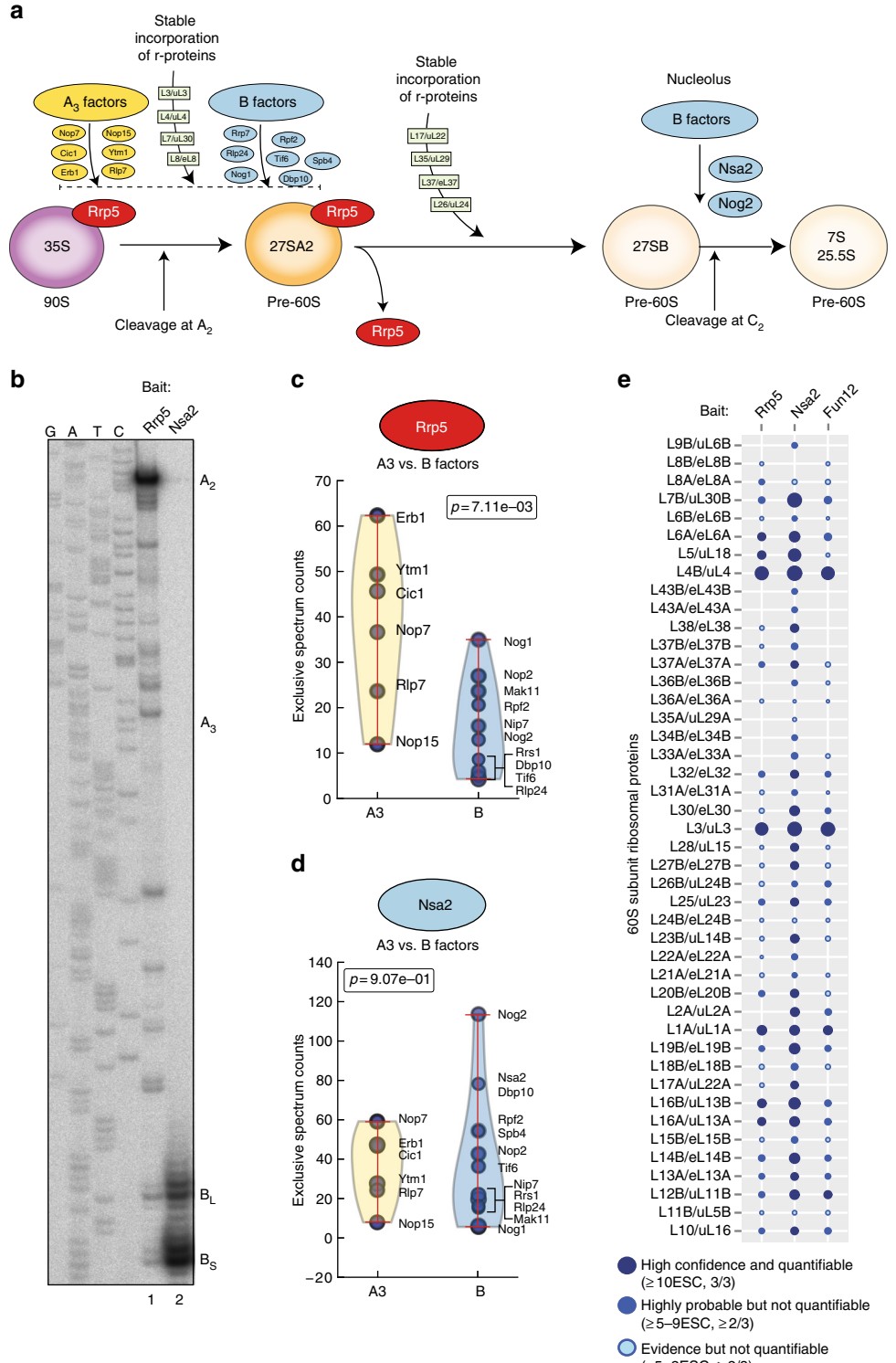

**Fig. 1** Affinity purification of 27SA$_2$ and 27SB pre-rRNA. **a** Schematic representation of the assembly steps of the nucleolar pre-60S particles. The 35S pre-rRNA is processed to 27SB pre-rRNA in various stages, which involves binding of assembly factors required for 27SA$_3$ processing (A$_3$ factors), assembly factors required for cleavage in ITS2 (B factors) and r-proteins. **b** Primer extension analysis of 27S pre-rRNAs isolated from the Rrp5-TAP and Nsa2-TAP pre-60S particles. The cleavage sites are indicated on the *right side* of the gel image. Lanes labeled with G, A, T and C represent dideoxy sequencing ladders. **c**, **d** *Violin plots* showing the mass spectrometry results for ribosome assembly factors. The plot shows the distribution of A$_3$ and B factors based on the ESCs from the Rrp5 (**c**) or Nsa2 (**d**) pre-60S complexes. The *p*-value was calculated using a *t*-test. **e** *Ball plots* showing the mass spectrometry results (ESC) for the r-protein composition of the complexes purified by Rrp5-TAP and Nsa2-TAP. The results were compared to the r-protein composition of the Fun12-TAP purified 80S particles. The larger the ball, the more enriched the protein was in the purification

In conclusion, the purified pre-60S complexes showed the expected protein composition, providing a good basis for our structure probing analyses.

**ChemModSeq analyses of purified pre-60S ribosomes.** To obtain pre-rRNA secondary structure information, the affinity purified pre-ribosomes were treated in vitro with the SHAPE chemical 1-methyl-7-nitroisatoic anhydride (1M7). Subsequently, the pre-rRNAs were resolved on agarose gels to remove contaminants and gel purified[15]. 1M7 generically reacts with all four nucleotides in flexible (mostly single-stranded) regions and acylates the 2′OH of the ribose[32]. To be able to assess the effect of protein binding events on pre-rRNA flexibility, we chemically probed pre-rRNAs from proteinase K (PK)-treated particles (Fig. 2a, Supplementary Fig. 1f). The rRNA regions bound by proteins can become more reactive to 1M7 in the deproteinized particles, in particular, those regions where proteins contact areas of single-stranded rRNA. This provided us with a unique approach to monitor the proposed gradual stabilization of r-protein association with pre-ribosomes[7, 31].

The purity and integrity of the 27S pre-rRNA samples were analyzed using a bioanalyzer (27S pre-rRNAs; Fig. 2a) or agarose gels (35S pre-rRNA; Supplementary Fig. 1b, lane 7). Primer extension was performed to assess whether the samples were sufficiently modified (data not shown). Subsequently, ChemModSeq libraries were prepared from these samples and high-throughput sequenced. ChemModSeq data from two biological replicates were pooled to increase the total coverage, followed by calculation of normalized 1M7 reactivities[33]. The 1M7 reactivity data is provided in Supplementary Data 2. Pearson correlation analysis revealed a positive correlation between 35S, $27SA_2$, and 27SB data (Fig. 2b–d). However, the 35S and $27SA_2$ data were considerably more highly correlated than either 35S or $27SA_2$ and 27SB. These results suggest that the 27S pre-rRNA undergoes major changes in RNA flexibility during the conversion of the $27SA_2$ to 27SB particle and that the pre-rRNA structure in the 35S and $27SA_2$ particles is quite similar. The highest correlation between any two data sets in our analysis was found for deproteinized $27SA_2$ and 27SB pre-rRNAs (Fig. 2e), indicating that many of the observed changes in reactivity between $27SA_2$ and 27SB may be due to RNA–protein interactions.

To determine how the structural information provided by the 1M7 probing experiments compares to the crystal structure of the 25S rRNA in the 60S subunit[34], for each domain in the 25S rRNA, we calculated area under ROC (receiver operating characteristic) curves using a binary classifier that only included nucleotides that were single stranded and not contacted by r-proteins in the crystal structure. As expected, these data indicate that the very early pre-60S intermediates analyzed in this study are structurally distinct from the mature 60S subunit (Supplementary Fig. 2). These data further indicate that domains I, II, and VI in the 35S and $27SA_2$ particles, and domains II, III, and V in the 27SB particle, are more similar to the crystal structure than other domains in these pre-60S particles.

**Gradual assembly of r-proteins into 60S pre-ribosomes.** To identify statistically significant differences in pre-rRNA nucleotide flexibility between the different particles, we employed the previously described ΔSHAPE algorithm[35]. SHAPE reactivities from different particles were subtracted and reactivities were averaged over a three-nucleotide sliding window to reduce local signal fluctuation. To be considered statistically significant, the difference in SHAPE reactivity (ΔSHAPE) had to be at least 1 standard deviation higher than the mean ΔSHAPE

values[35]. Because the 35S and $27SA_2$ data showed very similar SHAPE reactivity patterns, we focussed our analyses on the differences observed between 35S, $27SA_2$, and the 27SB data (Fig. 3a, b). The ΔSHAPE analyses confirmed that the 27SB particle showed a markedly different SHAPE reactivity profile compared to the other samples analyzed (Fig. 3b). Almost all of the 25S rRNA domains appeared to undergo conformational changes during the nucleolar stages of 60S assembly (Fig. 3a, b). The most striking observations were the increase in nucleotide flexibility in the 3′ end region of the 27SB pre-rRNA (domains IV–VI) and the fact that the 5.8S showed significantly higher SHAPE reactivity in the 35S and $27SA_2$ particles. The latter suggests that the 5.8S region undergoes major conformational changes during the conversion of $27SA_2$ to 27SB (also see below). In contrast, relatively few changes in flexibility were observed in ITS2 (Fig. 3b).

To be able to assess which changes in flexibility could be explained by protein-binding events, we next compared the pre-ribosome SHAPE data to the data obtained from the PK-treated samples (Fig. 3c). When we discuss PK-sensitive nucleotides, we refer to positions that showed significant ΔSHAPE values when comparing data from 35S or $27SA_2$ to 27SB particles, however, this change in flexibility was reversed in the PK-treated 27SB data. When we discuss PK-insensitive sites, we refer to positions that showed significant ΔSHAPE values between 35S-$27A_2$ and 27SB particles, which were not reversed by PK treatment of the 27SB particles. The majority of the nucleotides that showed a decrease in SHAPE reactivity in 27SB particles were PK-insensitive. These data suggest that these changes were unlikely the result of protein-binding events. The majority of these sites was located in domains 0–III (Fig. 3c), indicating that these domains fold into a more compact structure in the nucleolus. In contrast, most of the sites that showed increased flexibility in the 27SB particles were PK-sensitive and concentrated in domains III–V. These data suggest that RNA-binding proteins that interact with the 27SB particle (such as assembly factors or r-proteins) increase the flexibility of these domains. Interestingly, a fraction of these sites also overlapped with r-protein-binding sites (Fig. 3c; r-protein-binding sites, increase in flexibility).

For a number of nucleotides, we observed a decrease in flexibility in 27SB particles, and these changes were PK-sensitive (Fig. 3c; decrease in flexibility and PK-sensitive). We speculate that gradual assembly of r-proteins that bind with low affinity to early 60S pre-ribosomes, but become more stably bound in later intermediates, could explain some of these changes[3, 7, 24, 36, 37], although alternative explanations are possible (see Discussion). We refer to these sites as PKSRP (PK-sensitive SHAPE reactive sites bound by r-proteins). Interestingly, most of the PKSRP sites were mainly located/concentrated on the solvent-exposed interface (in particular in domain I) of the 60S subunit, but were under-represented on the subunit interface (which includes domains III and IV), the central protuberance, and the feet (Fig. 3d, Supplementary Data 3, Supplementary Movie 1).

**Formation of long-range inter-domain interactions.** Several of the PKSRP sites are contacted by the same ribosomal protein in the mature 60S subunit, but lie in different structural domains. Some of these nucleotides showed a decrease in reactivity during the conversion of 35S to $27SA_2$ and $27SA_2$ to 27SB. We interpret these changes as gradual stabilization of r-protein binding and formation or stabilization of long-range and/or inter-domain interactions. Some examples are discussed below and a complete overview can be found in Supplementary Data 3. The locations of the PKSRP sites suggest that interactions between domains

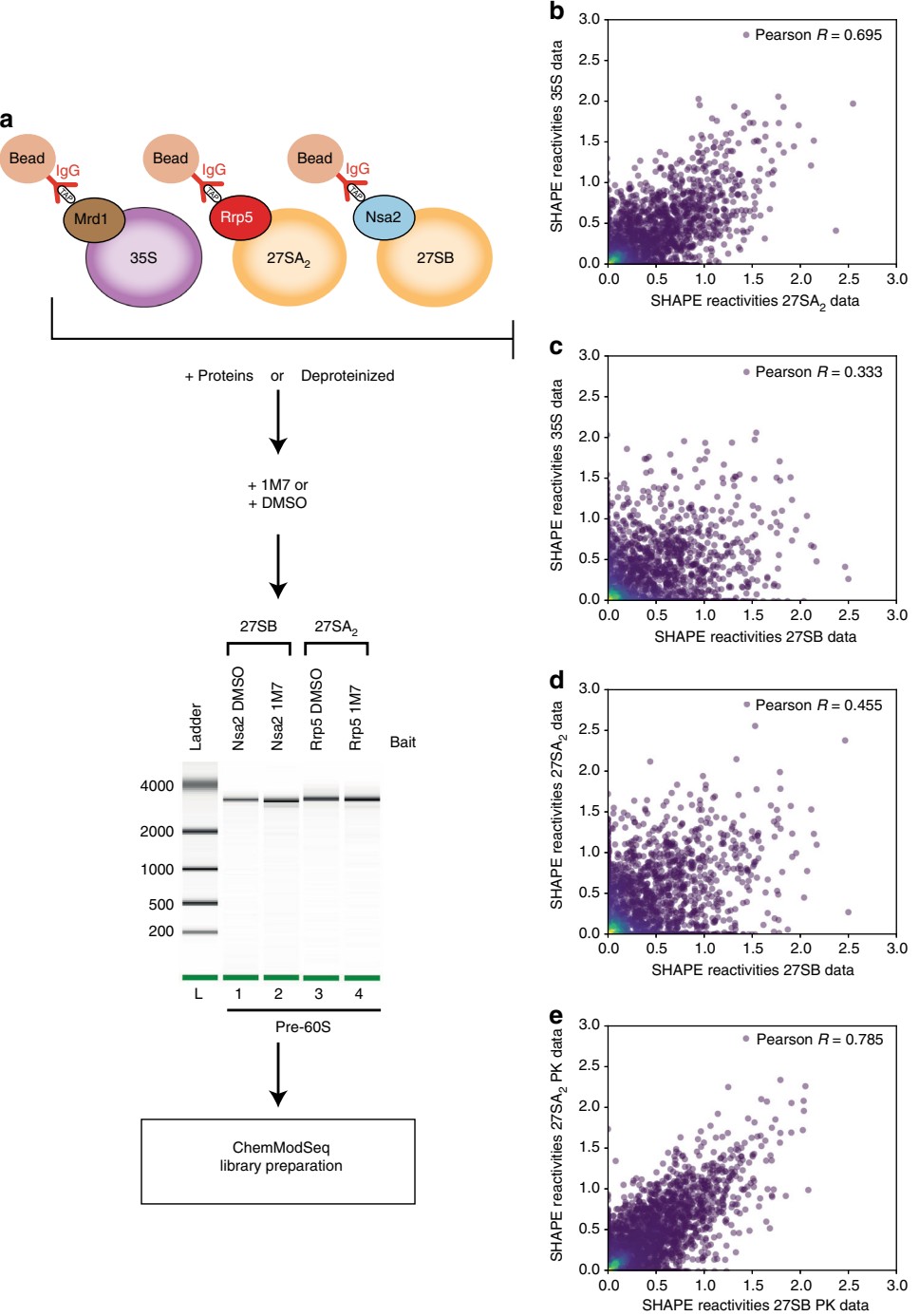

**Fig. 2** ChemModSeq experimental workflow and data analysis. **a** Particles containing the 35S, $27SA_2$, and 27SB pre-rRNAs were purified using TAP-tagged Mrd1, Rrp5, or Nsa2 as baits, respectively. Chemical probing was performed on the purified particles as well as PK-treated particles. As controls for natural primer extension stops, we also incubated purified particles and PK-treated particles with the 1M7 solvent (DMSO). The rRNAs were gel purified and their integrity was assessed on a bioanalyzer chip. **b**–**e** Many conformational changes take place between the $27A_2$ and 27SB particles. Shown are scatter plots that compare normalized SHAPE reactivity values of the 35S data to $27SA_2$ and 27SB data (**b**) and (**c**) and $27SA_2$ compared to 27SB data (**d**). Data generated from deproteinized particles is shown in **e**. The similarity between the data sets was determined by calculating Pearson correlation coefficients

I, II, V, and VI are formed by L4/uL4, L17/uL22, and L42/eL43 during the conversion of 35S to 27SB pre-rRNA (Fig. 4a). Similarly, we predict that several long-range interactions between domains III and V are formed by L2/uL2, L34/uL34, and L43/eL43 in the nucleolus (Fig. 4b). Interestingly, in the latter cases, the r-proteins initially bind domain V during the conversion of 35S to $27SA_2$ and then bind domain III in the 27SB particle.

Collectively, the data shown here indicate that rRNA-folding steps within many domains of the 25S rRNA region take place in the nucleolus, suggesting that many distant domains in the pre-rRNA could fold around the same time.

**Restructuring of 5.8S during the conversion of $27SA_2$ to 27SB.** The ΔSHAPE analyses as well as follow-up primer extension

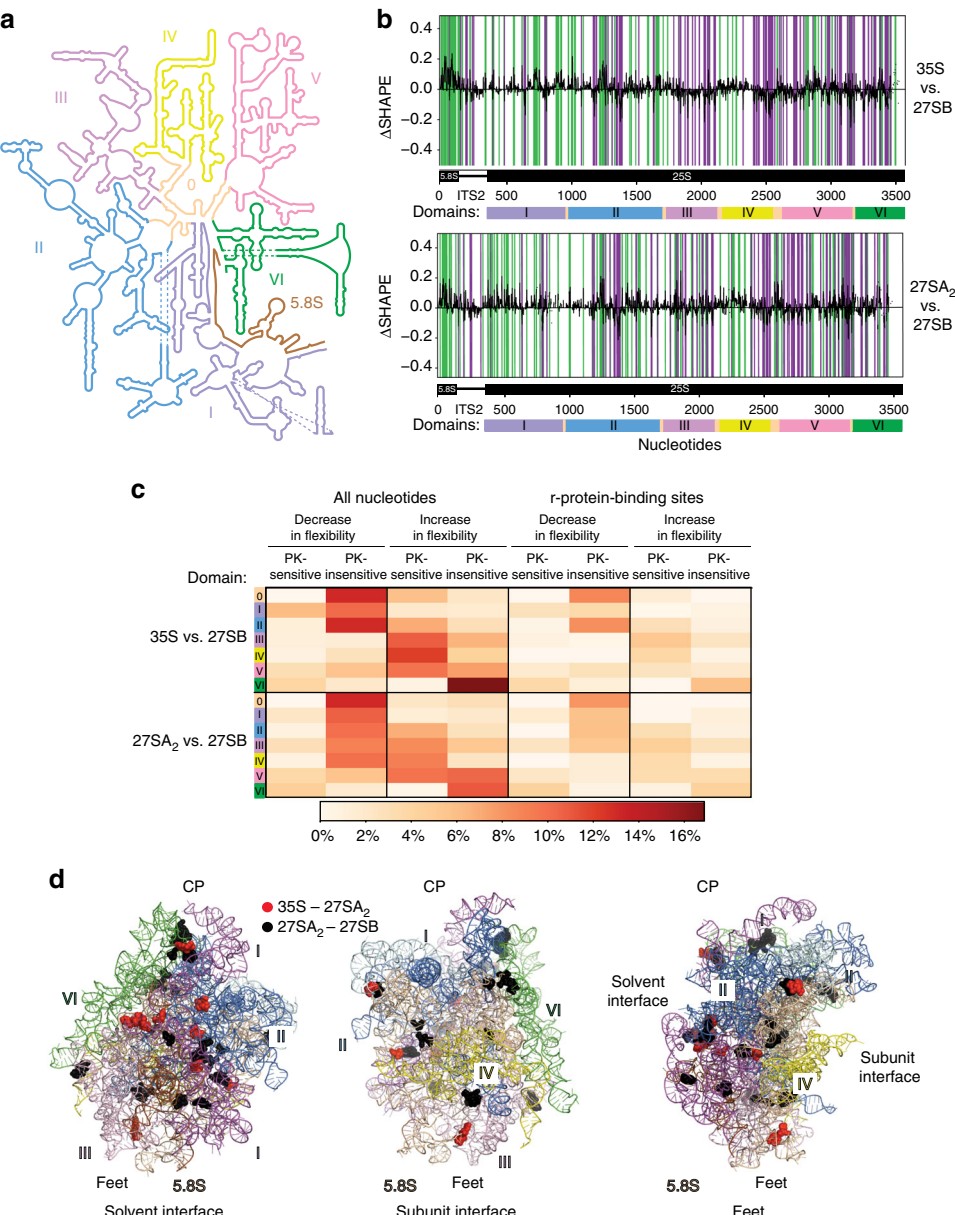

**Fig. 3** Folding of 25S rRNA domains. **a** Secondary structure model for the 25S and 5.8S rRNAs generated using RiboVision[64]. The individual domains are colored. **b** Differential SHAPE (ΔSHAPE) analysis of 35S, 27SA$_2$, and 27SB particles. The *histograms* show a comparison between 35S and 27SB data and 27SA$_2$ compared to 27SB data. If a region is highlighted in *green*, then it is more reactive in the 35S (*top*) or 27SA$_2$ (*bottom*) data. If a region is highlighted in *purple*, it is more reactive in the 27SB data. ΔSHAPE reactivities were calculated as using code developed by the Weeks lab[36]. **c** The *heat map* indicates the percentage of nucleotides in each 25S rRNA domain (indicated on the *left side* of each domain) that showed a decreased or an increase reactivity when comparing the various particles (35S vs. 27SB and 27SA$_2$ vs. 27SB) as well as whether these changes were sensitive or insensitive to PK treatment. Shown are the results for all the nucleotides as well as the nucleotides that overlapped with r-protein-binding sites in the 60S crystal structure[35]. **d** Overview of the PKSRP sites in the three-dimensional structure of the 25S rRNA[35]. Most of the proposed protein–rRNA interactions are concentrated on the solvent interface, with fewer PKSRP sites identified on the subunit interface, central protuberance (CP), and feet. Protein-binding sites were defined according to the crystal structure of the ribosome[35]

analyses indicated that the 5.8S region undergoes considerable structural rearrangements during the conversion of 35S to 27SB pre-rRNA (Fig. 3b, Supplementary Fig. 3). Closer inspection of modification patterns generated by both methods revealed that the major changes in RNA flexibility in this region occur specifically during the conversion of 27SA$_2$ to 27SB (Fig. 5a, compare lanes 2 and 3, Supplementary Fig. 3a–c). The observed restructuring likely coincides with binding of r-proteins to this region, which is indicated by the fact that some of

their nucleotide-binding sites in the 5.8S became less flexible in the 27SB pre-rRNA than in the earlier precursors (Fig. 5a; Supplementary Fig. 3), and that this reduction in flexibility was (at least in part) reversed in the deproteinized samples. Thus, our data suggest that the folding of some 5.8S regions may coincide with the binding of r-proteins L26/uL24, L35/uL29, and L37/eL37 to the 5.8S in the 27SB pre-rRNA.

Nop12 has previously been reported to be important for the formation of H5[38]. The Nop12-binding sites in 5.8S became less

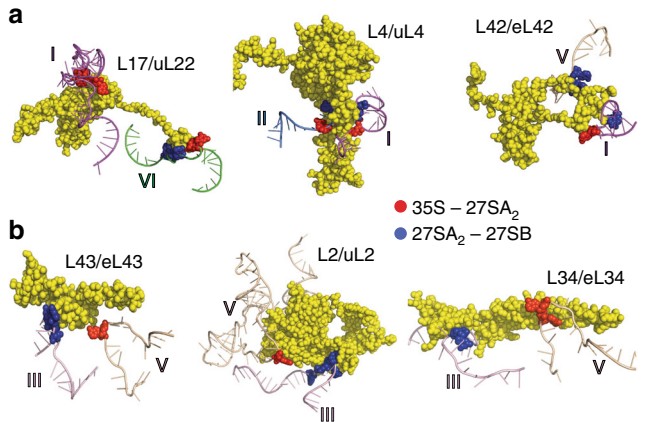

**Fig. 4** Models of inter-domain interactions formed during nucleolar stages of 60S assembly. The *yellow spheres* indicate r-proteins. Predicted protein–rRNA interactions formed during the conversion of 35S of 27SA$_2$ and the conversion of 27SA$_2$ to 27SB are indicated with *red* and *blue dots*, respectively. Protein-binding sites were defined according to the crystal structure of the ribosome[35]. **a** Binding of L17/uL22 L4/uL4 and L42/eL42 to domains I (*purple*), V (*wheat*), and VI (*green*) of the 25S during the conversion of 35S to 27SB pre-rRNA. **b** Sequential binding of L43/eL43, L2/uL2, and L34/eL34 brings domains III (*pink*) and V (*wheat*) in close proximity

flexible in the 27SB particles (Fig. 5a), but this flexibility was restored upon deproteinization. These data therefore suggest that Nop12-dependent formation of H5 may take place during the conversion of 27SA$_2$ to 27SB.

Helices H2, H3, parts of H4 and H10, which are formed by base-pairing interactions between 5.8S and the 25S (Fig. 5b), appeared to be mostly structured in all three precursors (Fig. 5a; Supplementary Data 2) and did not show any statistically significant differences in reactivity between the 35S and 27SB data (Fig. 5c, *black* nucleotides). These results indicate that many 5.8S–25S base-pairing interactions form early, possibly co-transcriptionally. Consistent with this idea, the 25S nucleotides that base-pair with these 5.8S regions generally reacted poorly with 1M7 and did not show significant changes in flexibility in all the particles analyzed (Fig. 5c, Supplementary Data 2). Thus, formation of 5.8S–25S base-pairing interactions precedes folding of 5.8S.

Other helices in 5.8S appeared to be unstructured in the 35S and 27SA$_2$ particles but became significantly less flexible in the 27SB particle. Nucleotides in H5–H8 were significantly more reactive to 1M7 in 35S and 27SA$_2$ particles compared to 27SB (Fig. 5c; Supplementary Data 2).

As we were unable to probe later pre-60S intermediates (see Discussion), we could not determine whether additional 5.8S restructuring occurs in later assembly intermediates. However, a recent cryo-EM study on a nuclear Nog2-associated pre-60S particle, which has a similar protein composition as the Nsa2 particle analyzed here, revealed that the structure of the 5.8S region is almost identical to that of 5.8S rRNA in the mature ribosome[12, 34] (Supplementary Fig. 4). Therefore, it is reasonable to conclude that in the Nsa2 particle, the 5.8S rRNA is almost completely folded and that therefore the majority of restructuring events takes place during the conversion of 27SA$_2$ to 27SB and involves formation of additional r-protein–rRNA interactions.

**ITS2 and flanking regions form compact structures**. Our structure probing data show that in contrast to the 5.8S region,

ITS2 is generally highly structured and does not appear to undergo any major changes in secondary structure during conversion of 35S to 27SB (Figs. 3b and 6b, c). We compared our data to three previously proposed ITS2 secondary structure models (Fig. 6a): the "ring" model centered on phylogenetic analysis[39], the "hairpin" model, largely based on enzymatic and chemical RNA structure probing[40], and a structure that we refer to as the "ring-pin model"[17]. The latter is mostly based on phylogenetic data and is a hybrid of the ring and hairpin models (Fig. 6a).

In general, our ChemModSeq data (Fig. 6b) and follow-up primer-extension data (Fig. 6c, Supplementary Fig. 5) agree best with the ring-pin structure, which we also found to be consistent with previously published mutagenesis data[41–43] (Supplementary Fig. 6).

The structure probing data support the proposed stem III in the ring-pin model, involving many A–U base-pairs (Fig. 6a–c). However, we note that although many nucleotides in region II are drawn as single-stranded nucleotides (Fig. 6a), it appears that they are less flexible compared to other loop regions within ITS2. A recent pre-60S cryo-EM structure uncovered the structure of three A$_3$ factors (Cic1, Rpl7, and Nop15) bound to fragments of ITS2[12], essentially covering almost completely regions I and II. This may explain why the probing data showed low flexibility in this region (Fig. 6d). We did observe a strong reduction in reactivity in the flexibility of U$_6$ near the 5′ end of ITS2 in the deproteinized data, reminiscent of what was observed in cells depleted of A$_3$ factors[44, 45] (also see Discussion). In the cryo-EM structure, this nucleotide appears to be contacted by Cic1 and Rlp7[12], apparently stabilizing a conformation that favors 1M7 modification. The observed reduction in 1M7 reactivity in deproteinized samples suggests that the high flexibility of this nucleotide is indeed associated with Cic1 and Rlp7 binding to the 5′ end of ITS2.

The 27SB particle reproducibly showed higher reactivities in ITS2 in the ChemModSeq data, particularly in the 100–200 region (Fig. 6b). Although this may indicate that ITS2 undergoes structural rearrangements in the Nsa2 particle, these sites were all located in bulge or loop regions (Fig. 6a), suggesting that these single-stranded regions are more effectively probed in the Nsa2 particle. This may be related to the differences in protein composition between the Nsa2 and Rrp5 particles.

## Discussion

Ribosome synthesis is a complex and dynamic process that involves the regulated folding of rRNA and hierarchical assembly of r-proteins. To obtain a step-by-step picture of the changes in rRNA folding during 60S maturation, we isolated the 35S, 27SA$_2$, and 27SB pre-rRNA intermediates from native pre-60S particles and measured nucleotide flexibilities using a high-throughput RNA structure probing method (ChemModSeq[15]). This allowed us to generate a global map of changes in rRNA flexibility during the early nucleolar stages of pre-60S maturation, providing novel insights into rRNA restructuring events as well as the establishment and timing of long-range interactions that take place during nucleolar stages of assembly. The work described here provides a solid basis for the future design of more focussed functional and structural studies.

We show that using specific TAP-tagged baits and gel purification, we can effectively separate the 35S, 27SA$_2$, and 27SB pre-rRNAs. This was supported by primer extension, bioanalyzer, and agarose gel analyses (Figs. 1b and 2a and Supplementary Fig. 1b), thus arguing for the presence of a single RNA species in each of the structural probing analyses. However, specific affinity purified pre-rRNA particles that contain a distinct

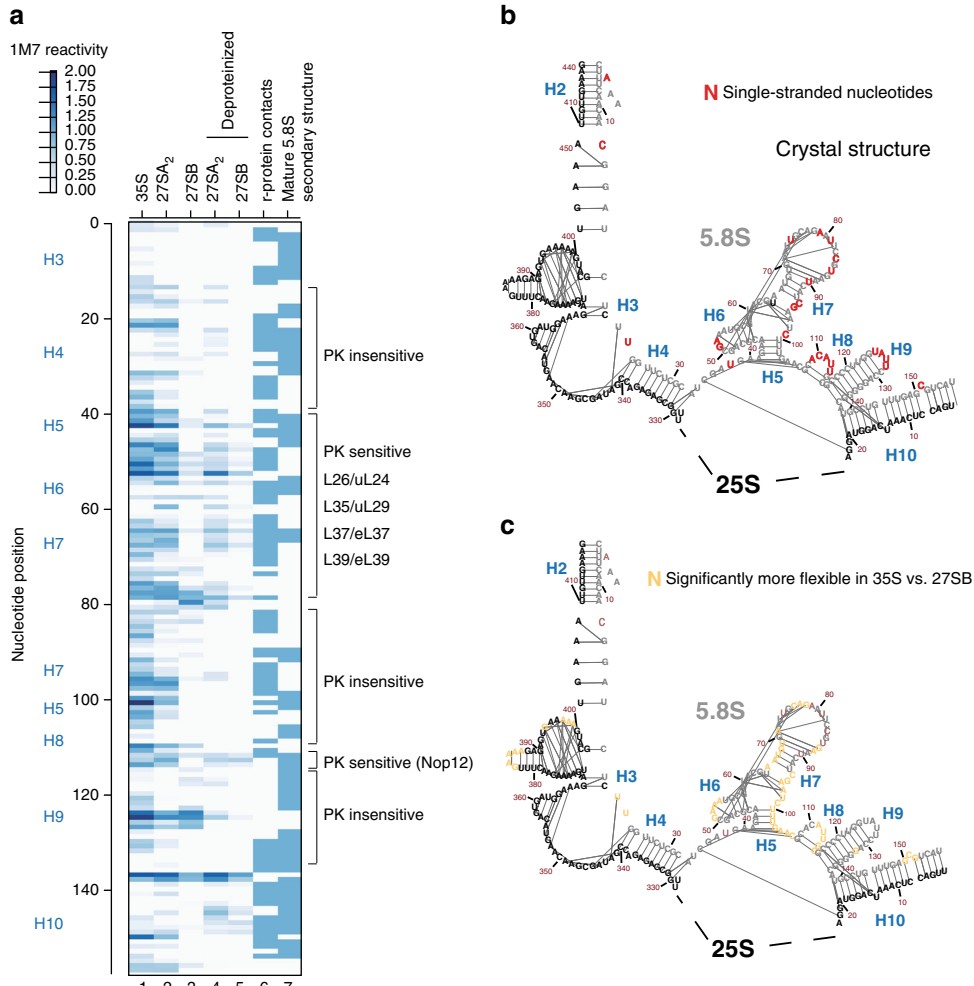

**Fig. 5** The 5.8S region undergoes a major restructuring event during the conversion of 27SA$_2$ to 27SB pre-rRNA. **a** Heat map representing the ChemModSeq 1M7 reactivities. Each *block* indicates a single nucleotide, the darker the block the higher the 1M7 reactivity. The *blue blocks* in lane 6 indicate r-protein-binding sites[35]. *Lane 7* represents the nucleotides that are base-paired (*blue blocks*) or single-stranded (*white blocks*) in the mature 5.8S rRNA crystal structure[35]. **b** Secondary structure of the 5.8S rRNA. *Black lines* indicate base-pairing interactions observed in the yeast 80S crystal structure. The *red* nucleotides are single-stranded in the crystal structure. **c** Overview of ΔSHAPE analysis in the 5.8S region. *Yellow-colored* nucleotides showed significantly higher SHAPE values in 35S compared to the 27SB particle data

pre-rRNA species could naturally differ in protein content and structure as these particles go through various assembly states. Thus, our structure probing data likely show averaged 1M7 reactivity profiles of these pre-rRNA particles. Nevertheless, our current data as well as our previously published work[15] clearly demonstrate that conformational changes can effectively be detected in purified pre-ribosomes using chemical probing. Most of the restructuring we detected here are likely rate-limiting events that are triggered by the release of ribosome assembly factors, binding of r-proteins, or the activity of energy-dependent enzymes.

We also wished to compare the pre-rRNA structures of nucleolar particles with later (nuclear) assembly intermediates. However, because the pre-rRNAs in these particles (5′-extended 25S species) could not be separated from contaminating 25S rRNA by gel electrophoresis, we were unable to perform these experiments. Generating sufficient quantities of highly pure 35S pre-rRNA for high-throughput RNA structure probing was very challenging. In order to obtain sufficient 35S pre-rRNA, we utilized a TAP-tagged Mrd1 mutant strain in which one of the RNA-binding domains was deleted (Mrd1Δ5-TAP). Although

pre-40S synthesis is impaired in the Mrd1Δ5 mutant, leading to the accumulation of 35S pre-rRNA, it remains associated with 90S pre-ribosomes and the 25S rRNA levels are similar to those of wild-type cells indicating that LSU processing is unaffected[20, 21] (Supplementary Fig. 1a, b). We therefore argue that the 35S pre-rRNA purified from the Mrd1Δ5 mutant can be efficiently processed into mature 25S and 5.8S rRNA. As 35S pre-rRNA accumulates in this strain, we cannot exclude that its folding kinetics differ from 35S pre-rRNA purified from a wild-type strain. However, we were able to purify several nanograms of 1M7-probed 35S pre-rRNA from the Rrp5-TAP strain sufficient for primer extension analysis (Supplementary Fig. 7). We did not observe any noticeable differences in reactivity pattern in the 5.8S region (the region where many changes in RNA flexibility were observed) between the 35S pre-rRNA purified from Rrp5-TAP and Mrd1Δ5-TAP. Therefore, we argue that the Mrd1Δ5 mutation does not have a significant impact on the folding kinetics of the 27S region of 35S pre-rRNA.

We also attempted in vivo structure probing experiments with the SHAPE chemical 2-methylnicotinic acid imidazolide (NAI)[46]. Although this worked well for the stable cytoplasmic pre-40S

complexes[15], much to our surprise, incubating the cells with NAI significantly reduced the recovery of the 35S and 27S pre-rRNA species with TAP-tagged baits. As a result, we were unable to generate sufficient NAI-probed pre-rRNA for high-throughput structure probing analyses. One

potential reason is that these early assembly intermediates become unstable during the relatively long incubation times (5–10 min) required for optimal NAI modification.

One could argue that ex vivo analyses of the pre-ribosomes do not fully capture the state of the particle under physiological

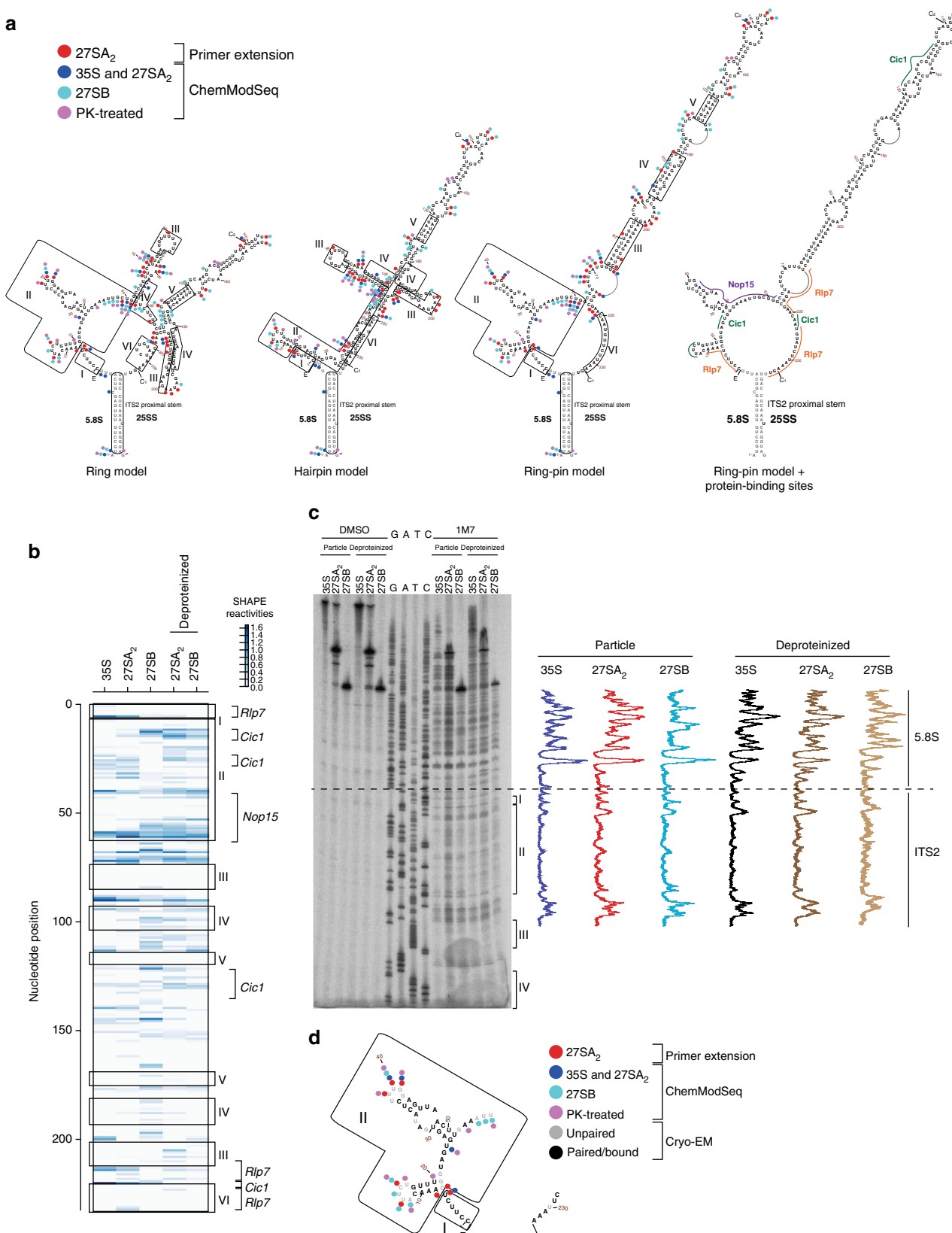

conditions. However, it is important to note that we purified the particles under conditions that are frequently used for tandem-affinity purification (TAP) of ribosome assembly intermediates as well as for cryo-EM studies. Therefore, the advantage of the ex vivo analyses is that we can directly compare our data with previously published structural and biochemical studies.

We report a detailed secondary structure analysis of the ITS2 region using the SHAPE reagent 1M7. In contrast to DMS used in previous studies[44, 45, 47, 48] that probes A's and C's, 1M7 allows to assay the flexibility of all four nucleotides. Our data indicate that ITS2 forms a very compact and helical structure, since 1M7 reactivity was generally much lower in ITS2 than in other regions, for example, 5.8S (Fig. 6c). Our data provides experimental support for a recently proposed secondary structure that we refer to as the ring-pin model, that is largely based on phylogenetic analyses[17]. Furthermore, our results are in good agreement with the recent cryo-EM structure of a late pre-60S intermediate in which two fragments of ITS2 (Fig. 6d) were resolved[12].

On the basis of the combined results of rRNA structure predictions, enzymatic and chemical probing experiments, and genetic analyses[37–40], it has been proposed that ITS2 may undergo a conformational switch that would allow 27SB pre-rRNA to be cleaved at site $C_2$. Our data provides no support for such a dramatic conformational change during nucleolar ribosome assembly, as ITS2 as well as its flanking regions that base-pair with 25S adopt a similar conformation in all particles analyzed (Fig. 6). However, we cannot exclude that such a switch occurs co-transcriptionally.

Chemical probing analyses have revealed that depletion of $A_3$ factors influence the DMS modification pattern of nucleotides near the 5′ end of ITS2[44, 47]. These data suggest that $A_3$ factors are involved in maintaining an open structure of ITS2 and that their release could re-structure ITS2, such as into the proposed compact hairpin structure. Our chemical probing data did not reveal dramatic differences in nucleotide flexibility of ITS2 upon deproteinization of the native $27SA_2$ and 27SB particles (Fig. 6). We did observe a significant reduction in the flexibility of $U_6$ in the 5′ ITS2 in our deproteinized samples, which appears to be contacted by Cic1 and Rlp7 in the cryo-EM structure. We therefore consider it more likely that the $A_3$ factors influence the ITS2 structure locally, rather than being involved in drastic structural rearrangements such as the proposed open (ring) to closed (hairpin) conformational switch. In agreement with this, the nucleotides that showed differential reactivity to DMS upon $A_3$ factor depletion[44, 47] overlap with or are in close proximity to Cic1- and Rlp7-binding sites[12, 44, 47]. These nucleotides did not strongly react with 1M7 in our particles or the PK-treated samples. A plausible explanation for these discrepancies is that 1M7 reacts with the 2′OH of the ribose, whereas DMS probes the bases of unpaired C's and A's. It is possible that, due to riboses being constrained by proteins binding at or near these nucleotides, they may not react with 1M7, whereas corresponding bases still can be modified by DMS.

Our analyses enabled us to determine the exact timing of major restructuring events in the 5.8S rRNA sequence. Formation of 5.8S–25S base-pairing interactions very likely takes place during transcription as those regions that form these base-pairing interactions generally reacted poorly to 1M7 and showed no significant changes in reactivity in the particles analyzed (Fig. 7). Moreover, while in the 35S and $27SA_2$ particles the 5.8S region appeared largely unfolded, it was highly structured in the 27SB-containing Nsa2 particles, suggesting that 5.8S undergoes major structural rearrangements during the conversion of $27SA_2$ to 27SB (Fig. 7). One possibility is that these restructuring events coincide with the binding of r-proteins to 5.8S. Our ChemModSeq data show that flexibility changes in some nucleotides are indeed PK-sensitive, hinting that this could be the case for some regions. The data do not allow us to determine what comes first: RNA folding followed by r-protein binding, or whether the r-proteins actually induce the folding of the RNA. However, the data do suggest that r-proteins are at least partially required to stabilize the more constrained structure of some nucleotides shortly after it has been formed. Consistent with this idea, deletion of L26/uL24 also increases the flexibility around helices H6 and H7 in the 5.8S rRNA[49]. Stable incorporation of r-proteins that interact with the 5.8S region (L26/uL24, L35/uL29, and L37/eL37) requires the $A_3$ factors, and it was proposed that rRNA remodeling by $A_3$ factors might generate the binding sites for these r-proteins, allowing their stable incorporation into the pre-60S[27]. $A_3$ proteins were abundantly detected in our $27SA_2$ and 27SB particles (Fig. 1). Yet depletion of Cic1 and Nop15 does not noticeably affect the in vivo structure of 5.8S[44], suggesting that $A_3$ factors do not directly influence 5.8S folding.

L17/uL22, L35/uL29, and L37/eL37 were recently shown to be essential for the association of the late B-factors[48]. Thus, our current working hypothesis is that the stable incorporation of these 5.8S r-proteins into pre-60S complexes signals that 5.8S rRNA folding has been successfully completed, paving the way for the assembly of the late B-factors and cleavage at site $C_2$ (Fig. 7).

Recent work has indicated that many r-proteins bind early pre-60S complexes with low affinity and that their association gradually becomes more stable during later assembly steps. Interestingly, we frequently observed a decrease in flexibility of nucleotides that overlapped with r-protein-binding sites identified in the 60S crystal structure[34]. The change in flexibility of these nucleotides was (largely) restored when the particles were deproteinized (PK-sensitive sites; PKSRP), indicating that the change in nucleotide flexibility was the result of protein–RNA interactions. These data therefore allowed us to speculate about the formation of r-protein–rRNA interactions during pre-60S assembly. The chemical probing data do not provide direct evidence that the observed changes in rRNA flexibility were indeed mediated by r-protein binding to the nucleotide. Indeed, there are a few alternative scenarios that could explain the appearance of PK-sensitive sites. It is possible that r-proteins binding in the vicinity or transient binding of ribosome assembly factors could

**Fig. 6** ITS2 forms a very compact structure. **a** Overview of the various proposed secondary structure models for ITS2[17, 40, 41]. Roman numerals indicate the regions discussed in the main text. *Dots* of different colors represent nucleotides with SHAPE reactivity ≥0.4 in ITS2 primer extension data (*red*) and 35S and $27SA_2$ particles (*dark blue*), 27SB particles (*light blue*), and deproteinized samples (*magenta*) based on ChemModSeq data. Note that the primer extension results for the 3′ end are not shown as this region showed high variability in reactivity. The known binding sites for assembly factors that interact with ITS2 are highlighted in the ring-pin model. **b** Heat map representing the ChemModSeq 1M7 reactivities. Each *block* indicates a single nucleotide. The darker the block the higher the 1M7 reactivity. **c** Primer extension analysis of the 5′ end of ITS2 and the 3′ end of 5.8S region and line scans of the signal intensities of each lane. **d** RNA structure probing data agrees well with the cryo-EM structure of ITS2 fragments. Shown is the secondary structure of parts of ITS2 that were resolved by cryo-EM[12]. The *black* nucleotides in the sequence are nucleotides that are either base-paired or bound by assembly factors in the structure. *Gray* nucleotides are predicted to be single-stranded as they are not involved in Watson–Crick or Hoogsteen base-pairing interactions and are not predicted to be bound by proteins

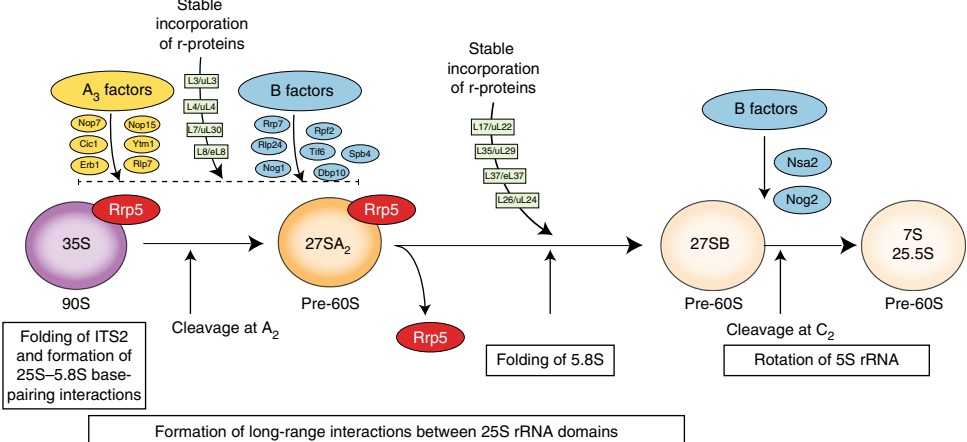

**Fig. 7** Model of rRNA structural changes and pre-60S remodeling steps that take place during the nucleolar stages of 60S synthesis. ITS2 forms a highly compact structure in the nascent transcript. We predict that base-pairing interactions between 25S and 5.8S have already been formed co-transcriptionally. Major rRNA restructuring events occur in pre-ribosomal complexes during the conversion of 27SA$_2$ pre-rRNA to 27SB pre-rRNA. These include the compaction of the 5.8S region and formation of long-range interactions in the 25S sequence. All these changes occur after Rrp5 dissociates and in the presence of A$_3$ and early B factors. After C$_2$ cleavage, the 5S rRNA rotates to adopt its final structure[64]

induce the observed change in flexibility. For example, we observed changes in the binding sites for L42/eL42, which we were unable to detect by MS and which were found to be mostly enriched in the late (Arx1) pre-60S complexes and the mature 80S ribosomes[36]. Therefore, it is plausible that, in some cases, the observed changes in flexibility could be the result of another protein (such as an assembly factor) occupying these sites.

It has been suggested that assembly of pre-60S particles takes place in an hierarchical fashion and that the structural domains are formed in 5′–3′ directionality (or at least some regions in the 5′ domain need to be assembled before the formation of 3′ domains can take place)[50]. This model proposed that the convex solvent side of the 60S starts to form first followed by the polypeptide exit tunnel and finishing with the central protuberance. A large number of r-proteins interact with multiple domains of the 25S rRNA[34] and therefore, intuitively, it seems logical that binding of r-proteins must affect the folding of multiple domains. It is conceivable that long-range interactions could already be established at a very early stage of assembly and that several (parallel) assembly pathways might exist. Our analysis provided an opportunity to address this from the RNA perspective. By focussing on those changes in nucleotides that we predicted were the result of r-protein–rRNA interactions (PKSRP sites), we asked which domains of the 25S rRNA were most likely to undergo r-protein induced folding. Consistent with the proposed hierarchical assembly model[50], our results do indicate that most of the PKSRP sites in nucleolar pre-60S complexes concentrated on the solvent interface of the 60S subunit (domains I and II), whereas fewer predicted changes in protein–rRNA interactions were detected on the subunit interface side (domains III and IV) and the central protuberance, which were proposed to fold later (Fig. 3c). However, our results also support the idea that r-proteins forge contacts between many distant domains (such as interactions between domains I, II, III and V, VI) at a very early stage.

The next challenges will be to determine what role assembly factors play in directing these folding steps, and to generate detailed maps of the 60S rRNA-folding pathways. We expect that (near) atomic resolution cryo-EM, combined with RNA structure probing and crystallographic data, will greatly accelerate our understanding of the dynamics of ribosome assembly.

## Methods

**Yeast strains**. *S. cerevisiae* strain BY4741 (*MATa*; *his3Δ1*; *leu2Δ0*; *met15Δ0*; *ura3Δ0*) was used as the parental strain. The TAP-tagged Rrp5 and Nsa2 strains were taken from the yeast TAP-fusion open reading frame (ORF) collection (GE healthcare) and were grown in YPDA media (2% glucose, 1% yeast extract, and 2% peptone) (Formedium). The TAP-tagged Mrd1 mutant was previously described[51] as ASY055. ASY055 cells were grown overnight in YPGA media (1% yeast extract, 2% peptone, 2% galactose, and 2% raffinose), shifted to YPDA media, and grown for 8 h at 30 °C to an optical density (OD$_{600}$) of 1.

**Synthesis of 1M7**. Synthesis of 1M7 was performed as previously described[32, 52]. Briefly, sodium hydride (60% dispersion in mineral oil, 1.4 equiv, 5.72 mmol, 229 mg) was suspended in 13 ml of dimethylformamide (DMF anhydrous grade from Sigma-Aldrich) under a nitrogen atmosphere and stirred for 5 min. 4-Nitroisatoic anhydride (1 equiv, 4.08 mmol, 850 mg) in DMF (9 ml) and then methyl iodide (1.05 equiv, 4.29 mmol, 609 mg, 270 µl) were added dropwise, and the reaction stirred for 24 h. The reaction mixture was added to partially frozen 1 M HCl (150 ml) forming a yellow precipitate. This was recovered by filtration, washed with water, then ether and dried under vacuum for 14 h to yield crude product. This was then triturated from ether/pentane and dried under vacuum for a further 14 h to yield 1-methyl-7-nitroistoic anhydride (585 mg, 64%; Supplementary Data 4).

**Immunoprecipitation and chemical modification of rRNA**. Cells were collected at OD$_{600}$ ~ 1.0 and lysed in 4 V g$^{-1}$ ice cold TMN150 buffer (50 mM Tris pH 8, 150 mM NaCl, 0.1% NP-40, 1.5 mM MgCl$_2$, and Roche protease inhibitor cocktail) by vortexing for 5 min with 3 V per g of 0.5 mm Zirkonia beads (Thistle)[53]. The lysate was incubated with 400 µl IgG Sepharose beads (Amersham) for 1 h at 4 °C. The beads were then washed three times 5 min with 10 ml TMN150 buffer at 4 °C and finally resuspended in 200 µl TMN150 buffer. Subsequently, the beads were incubated with 1M7 (dissolved in DMSO, 12.5 mM final) or DMSO (negative control) for 3 min at room temperature. The final concentration of DMSO was 5%. To deproteinize the particles, complexes bound to IgG beads were resuspended in 200 µl TMN150 buffer containing 1% SDS and incubated with 2 µg per µl PK for 3 h at 14 °C prior to 1M7 modification. The RNA was phenol–chloroform-extracted, ethanol-precipitated, and resolved on a 1% low-melting agarose gel (Seaplaque GTG agarose, Lonza). The pre-rRNA were subsequently excised from the gel and extracted by melting the agarose melted at 65 °C with five gel volumes of extraction buffer (10 mM Tris-HCl, 1 mM EDTA, pH 7.5). RNA was subsequently purified by phenol–chloroform extraction and ethanol-precipitated. Purified 27S pre-rRNAs were quantified on a 2100 bioanalyzer (Agilent) using an RNA Nano 6000 assay. All experiments were performed at least twice, and for each particle, we included separate control experiments. This was performed to minimize variability in coverage profiles between modified and control experiments.

**Primer extension analyses**. Reverse transcription was performed using Superscript III (Invitrogen) with 8 ng of purified rRNA and $^{32}$P-radiolabeled oligonucleotides. Samples were incubated at 45 °C for 45 min, after which 1 µl of Exonuclease I (NEB) was added, followed by a 30-min incubation at 37 °C.

Reactions were stopped by the addition of 5 µl of RNA loading dye (NEB). Sequencing ladders were prepared using a plasmid containing an rDNA repeat (pBS-rDNA) and Sequenase v2.0 (USB/Affymetrix). Samples were resolved on a 6% polyacrylamide/8 M urea gels. Band intensities were quantified using the FLA5100 phosphoimager system (Fuji). Signal decay correction was done using exponential function ($b = 0.999$ for DMSO and $b = 0.995$ for 1M7 treated samples). Structural reactivities were calculated by subtracting band intensity in the control lane from that of corresponding nucleotide in 1M7 lane. All negative values were set to zero. All values were 2–8% normalized. 1M7/DMSO ratios were calculated and reactivities of nucleotides with 1M7/DMSO < 1.5 were also set to zero to avoid false-positives.

Oligonucleotides used for primer extensions were:

ITS2_143-162 (5′-GATTAGCCGCAGTTGGTAAA-3′), ITS2_58_77 (5′-CATCCAATGAAAAGGCCAGC-3′), ITS2nt162 (5′-GATTAGCCGCAGTTGGTAAA-3′), and 25Snt47 (5′-GATATGCTTAAGTTCAGC-3′).

**Northern blot analysis**. Northern blot analysis was performed essentially as described[54], using the following oligonucleotides:

RDN1c (5′-ATGAAAACTCCACAGTG-3′)
RDN1e (5′-GGCCAGCAATTTCAAGT-3′)

**ChemModSeq library preparation and sequence data analysis**. ChemModSeq libraries were prepared as described by Hector et al.[15] A detailed overview of all the adapter sequences and primers used is provided in Supplementary Table 1. Individual libraries were pooled based on concentration and barcoding, and paired-end sequencing was performed on HiSeq systems by Edinburgh Genomics (Edinburgh). Raw data processing was carried out using the pyCRAC software package[55] and our ChemModSeq pipeline (see Data and code availability section below). Data from replicate experiments were summed to increase the total coverage. The ChemModSeqPipeline python script runs the complete data processing pipeline. We initially used the TCP_EM algorithm[15] for identifying 1M7-modified nucleotides. However, our recent comparative analyses showed that, although this algorithm has a high specificity, it generally calls a much lower number of nucleotides modified compared to our SHAPE reactivity script[56], which is why we decided not to use it for these analyses. In this manuscript, we calculated SHAPE reactivities according to the StructureFold method[33] using the CalculateSHAPE_reactivities.py script.

The ChemModSeq pipeline first trims the adapter sequences from the reads using Flexbar[57] followed by demultiplexing of the samples using pyBarcodeFilter from the pyCRAC package. This removed the barcode information in the forward reads except for the last random nucleotide, which was removed using the TrimNucs.py script. The random barcode information (seven random nucleotides) was subsequently used to collapse the data (pyFastqDuplicateRemover.py). The resulting fasta output files were then mapped to the yeast rDNA sequence using novoalign 2.04. The pyReadCounter.py script was subsequently used to generate gene transfer format (GTF) interval files from the novo files that contained merged coordinates of read pairs that mapped to the 35S within 1000 nucleotides from each other. Only properly paired reads were considered for the analyses. The resulting GTF files were then used to generate pileups containing drop-off counts and coverage information for the 35S and 27S sequences (using pyGTF2sgr.py). These count data were subsequently used to calculate SHAPE reactivities using the CalculateSHAPE_reactivities.py script, which also applies 2–8% normalization to scale the values between 0 and ~2. The ΔSHAPE reactivities were calculated using python code from the Weeks lab[35]. The ΔSHAPE values for the 27SB region in each sample analyzed is provided in Supplementary Data 2.

**Mass spectrometry**. Rrp5p-ProtA, Nsa2-ProtA, Fun12-ProtA, and controls (W303a, ProtA-tag expressed alone) were grown in YPD, collected in logarithmic phase ($OD_{600}$ 0.6–0.85), and cryo-lysed as previously described[58]. The affinity purifications were performed in technical triplicates for samples and singulate for controls. The purification of Rrp5-ProtA, Nsa2-ProtA was performed in RNP150 (20 mM HEPES-KOH pH 7.4, 110 mM KOAc, 0.5% Triton X-100, 0.1% Tween-20, 1 : 100 solution P, 1 : 5000 antifoam A, and 150 mM NaCl) and purification of Fun12-PrA was performed in TBT150 (same as RNP150 supplemented with 2 mM $MgCl_2$, 1 : 1000 DTT) as described elsewhere[58].

Briefly, 0.5 g of slightly thawed, cryo-lysed cell powder was vortex-resuspended in nine volumes of extraction buffer and further homogenized using a polytron (30″ on ice). The centrifuged-cleared sample ($3200 \times g$, 10′, 4 °C) was incubated (30′, slow rotation, 4 °C) with 7.5 mg of Dynabeads conjugated with rabbit IgG (160 µg IgG per mg bead) pre-equilibrated in extraction buffer. Prior to on-bead digestion, the beads were washed 10 times with extraction buffer; once with 100 mM $NH_4OAc$/0.1 mM $MgCl_2$/0.1% Tween-20 (room temperature, slow rotation, 5′); four times with 100 mM $NH_4OAc$/0.1 mM $MgCl_2$ (three times fast and once at room temperature, slow rotation, 5 min); and once with 20 mM Tris-HCl (pH 8.0). Subsequent to single-step affinity purification (ssAP) purification, one-tenth of each affinity purification sample was eluted (0.5 M $NH_4OH$) and resolved on 4–12% gradient Bis-Tris NuPAGE gels for silver staining. The remaining samples were on-bead trypsin digested in a volume of 50 µl (20 µg per ml trypsin (Sigma, proteomics grade) in 20 mM Tris-HCl (pH 8.0), 37 °C, 900 r.p.m., 16–20 h, and

stopped with 2% formic acid[59] and analyzed by tandem MS as described previously[58].

Briefly, dried tryptic digests were zip tipped (Millipore, as per supplier recommendations) and loaded onto a C-18 reverse-phase capillary column (15 cm × 75 µm (Length (L) × internal diameter (i.d.)); 5 µm particles, 300 Å) for electrospray ionization time-of-flight MS. Ionization was performed by a Proxeon nanoelectrospray Flex ion source set to 1.3-1.7 kV. Chromatographic separation of peptides was performed at 250 °C on an Easy-nLC II system (Proxeon Biosystems) at 300 nl per min, first over a gradient (20 min) from 95% solvent A (water—0.1% formic acid) to 25% B (100 % acetonitrile, 0.1% formic acid); then to 45% B (40 min); and finally to 80% B (10 min). Parent ion scans were performed in the HCD cell of the Q-Exactive (Thermo Fisher Scientific) over a mass range of 360–2000 m/z at a resolving power of 70 000. The automatic gain control (AGC) target and maximum ion fill time (IT) were set to $1 \times 10^6$ and 100 ms respectively. Fragments were obtained with a normalized collision energy of 27, an intensity threshold of $1.2 \times 10^4$, and underfill of 0.9%. Data-dependent mode was used, in which 14 most intense precursor ions were isolated with a dynamic exclusion window of 15 s. MS/MS was acquired using a resolution of 17 500, an AGC target of $1 \times 10^5$, and a maximum IT of 50 ms. Data were processed with Thermo Excalibur to generate a raw file. Mascot search server (version 2.3.02[60]) was used with a parent tolerance of 10 p.p.m. for precursor ions, 0.52 Da for fragments, and only considering one possible missed cleavage as well as a mass change of +16 for methionine oxidations in the mass calculation. Data were searched against S. cerevisiae database of NCBI (txid4932, 20140813 release) and analyzed in Scaffold (version 3.6.4). The threshold and false discovery rates were set to 80% and 0.37%, respectively, for peptides, and to 95% (1 peptide minimum) and 1.9%, respectively, for proteins.

Exclusive spectrum counts (ESCs) were used for analysis to discriminate against peptides shared between paralog proteins. For each prey, the highest values obtained in the controls were removed from those of the samples during analysis. Preys were defined as legitimate if they had at least two significantly enriched spectrum counts in at least two of the three replicates and selected for semi-quantitative analysis. The preys not selected for analysis are listed in Supplementary Data 1. For semi-quantitative analysis, exclusive spectral counts for L3 were used to normalize the data because it enters early during ribosome biogenesis and it was found in the three bait proteins in similar counts[50, 61, 62]. Semi-quantitative results were color-coded into four categories, high confidence and quantifiable (in red, >10ESC, 3/3); highly probable but not quantifiable (in orange, 7.5–9.9ESC, in 2/3); probable but not quantifiable (in yellow, 5–7.49ESC, in 2/3); and some evidence (in gray, 2–4.9ESC, 2/3). Standard deviations were calculated for the triplicates. Results are shown in alphabetical order and by categories (Supplementary Data 1).

**Code availability**. The scripts used for processing these data, a list of dependencies as well as a pdf file explaining our library design can be found on bitbucket.org/sgrann/chemmodseqpipeline.

**Data availability**. The ChemModSeq sequencing data is available from the Gene Expression Omnibus under accession number GSE83821. The data that support the findings of this study are available from the corresponding author upon request.

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

## Acknowledgements

This work was supported by grants from the Wellcome Trust to S.G. (091549) to E.B. (096997) and the Wellcome Trust Centre for Cell Biology core grant (092076). F.L. was supported by a short-term EMBO fellowship (ASTF 568-2012). L.W., S.B., and F.L. are supported by the Swedish Research Council. M.O. is supported by a Discovery Grant from the Natural Sciences and Engineering Research Council of Canada (RGPIN 386315-MO) and a Project Grant from Canadian Institutes of Health Research (PTJ-153313). N.D.-W. is supported by the Engineering and Physical Sciences Research Council. S.L.C. acknowledges support from a European Research Council Starting Grant (336935). Next-generation sequencing was carried out by Edinburgh Genomics, The University of Edinburgh. Edinburgh Genomics is partly

supported through core grants from NERC (R8/H10/56), MRC (MR/K001744/1), and BBSRC (BB/J004243/1).

## Author contributions

E.B., L.W., M.O., and S.G. designed the research. E.B., S.G., F.L., L.-C.A., S.B., R.v.N., and R.D.H. performed the experiments and analyzed the data. S.B. quantified the primer extension data and C.T. compared the probing data to cryo-EM and crystal structures. N.D.-W. and S.L.C. synthesized the 1M7. L.-C.A. and M.O. performed the mass spectrometry experiments and data analyses. All authors contributed to the writing of the manuscript.

## Additional information

**Competing interests:** The authors declare no competing financial interests.

