## [Peer Review File · Nature Communications]

Reviewers' comments:

Reviewer #1 (Remarks to the Author):

Burlaco et al. analyze in the submitted manuscript changes in the folding states of rRNA precursors during early steps of large ribosomal subunit maturation in the yeast *S. cerevisiae*. They affinity purified from yeast cells three different early to intermediate large ribosomal subunit precursor populations and probed them in vitro with the agent IM7 to determine changes in conformational flexibility of individual pre-rRNA nucleotides. The obtained dataset represents an important resource for people interested in early yeast large ribosomal subunit maturation and provides new insights into this process!

Specific comments and questions (in order of appearance of relevant issues in the manuscript):
Introduction, p.4: "To generate an overview pre-rRNA folding steps during the early stages of 60S maturation on a single nucleotide resolution level ...". Phrasing should be corrected.

Results: In general, my impression was that the manuscript would benefit from largely shortening the results section and focussing there on a neutral description of experimental strategies applied and the results obtained. Other parts of the results section (see below), which mainly provide possible interpretation of the data and of relevant literature, could be extended and moved to the discussion section.

Results, paragraph entitled "Purification and characterization of pre-60S intermediates for RNA structure probing": The authors use a *mrd1-dRBD5* mutant strain in which 35S pre-rRNA processing is delayed to facilitate efficient purification of early 35S containing pre-ribosomes. They compare conformational flexibility of rRNA nucleotides in the resulting particles with the ones in more matured large subunit precursors. It is unclear how much of the detected changes can be attributed to the mutant situation and how much reflect changes in pre-rRNA folding happening during the 35S to 27SA transition in a wildtype situation. This point should be communicated to the reader to avoid misinterpretation of the presented dataset.

Results, paragraph entitled "Purification and characterization of pre-60S intermediates for RNA structure probing": Is there a reason that the authors characterized the RNA and protein content of Rrp and Nsa2 particles but not of *mrd1-dRBD5* particles (see Figure 1b-d)?

Results, paragraph entitled "Purification and characterization of pre-60S intermediates for RNA structure probing": In Fig. 1e the result of the analysis of the ribosomal protein content of Fun12 associated ribosomal particles is shown. The reason for performing this experiment is not explained in the text and I could also not find any other reference to this experiment in the results or discussion sections. I guess the Fun12 particles should reflect (more) mature(d) 60S subunits. In this regard, why do they seem depleted of several ribosomal proteins which were detected in precursor particles (L9, L43, L38, L35, L34 and others)?

Results, paragraph entitled "Purification and characterization of pre-60S intermediates for RNA structure probing": Is there a reason that different buffers were used for affinity purification of particles for proteome analysis (buffer RNP150 and buffer TBT150) when compared to buffers used for purification of particles used for RNA structure probing experiments (buffer TMN150)?

Results, paragraph entitled "Purification and characterization of pre-60S intermediates for RNA structure probing": The authors did not include in their analysis large subunit precursors downstream of 27SB pre-rRNA. This might have further supported their notion that significant aspects of large

ribosomal subunit pre-rRNA folding occur at early stages of maturation. It could have also served as further control for their approach and might have substantiated several aspects of the interpretation of their results. The authors might explain to the readers their choice to focus on the analysis of early 60S precursors.

Results, paragraph entitled "ChemModSeq analyses of purified pre-60S ribosomes": The authors probe nucleotide flexibility in ex vivo isolated pre-ribosomal populations treated or not with proteinase K. They use differences in nucleotide flexibility between these two conditions to distinguish "rRNA folding events" and "protein binding events" (see for example Fig. 2b). Proteinase K treatment was done, according to the reference cited, for 3 hours at 14°C. Did the authors test whether protein digest in these conditions was complete or whether partially protected parts of proteins remained associated with rRNA with possible influence on its structure? The authors classify "rRNA folding events" as the ones for which changes in nucleotide flexibility from earlier to later pre-ribosomal populations was observed which resisted proteinase K treatment. That criterion seems quite disputable. Beside technical issues, as partial proteinase K digest (see above), the authors themselves discuss in this paragraph scenarios leading to false classification, as transient protein-RNA interactions with lasting impact on RNA fold. I would much prefer neutral terms to describe these classes, as "proteinase k sensitive" and "proteinase k insensitive" together with an appropriate discussion of their possible interpretation in the discussion section. That could also help to shorten this quite lengthy part of the results section. One could briefly introduce the proteinase k experiment as an approach to detect protein dependent rRNA folding states and come back to it in the discussion section. The following section listing all possible combinatorial categories together with some (but not all) possible interpretations could then be drastically shortened.

Results, paragraph entitled "ChemModSeq analyses of purified pre-60S ribosomes": The following sentence contains a duplication: "The fact that the 1M7 reactivity does not (dramatically) change when deproteinizing the samples, however, suggests that the formed structure does not require the presence of the proteins to remain stable, suggested by the fact that the 1M7 reactivity does not (dramatically) change when deproteinizing the samples."

Results, paragraph entitled "ChemModSeq analyses of purified pre-60S ribosomes": Supplementary table S2, containing 1M7 reactivities per nucleotide, would benefit from a more detailed legend. That is important if the dataset should serve as a resource for the community.

Results, paragraph entitled "ChemModSeq analyses of purified pre-60S ribosomes": The figure legend of figure 2 a) needs rephrasing.

Results, paragraph entitled "ChemModSeq analyses of purified pre-60S ribosomes": "strikingly, 40% of the analyzed nucleotides .." and Introduction p.4: "Our analyses suggest that roughly one third of 60S pre-rRNA folding occurs cotranscriptionally..". In the results part the authors argue that 40% of all the analysed nucleotides did not show any changes in 1M7 reactivity in the pre-rRNA populations analyzed and, in their opinion, that suggest that a significant fraction of the 27S pre-rRNA already folds co-transcriptionally (results page 10). Do we know the amount of changes in conformational flexibility happening in pre-ribosomal populations downstream of the ones analyzed here? And how do the authors quantify changes in "60S pre-rRNA folding" states which include changes in the tertiary structure exemplified by drastic rotations of parts of the central protuberance observed at later stages of 60S maturation? The authors use moreover a mutant in which pre-rRNA processing is delayed to be able to purify efficiently the earliest (posttranscriptional) LSU precursor included in their analysis (see also discussion below). This delay in pre-rRNA processing obviously might increase the time for pre-rRNA to fold in early pre-ribosomal populations. Therefore, and due to various other reasons in a wildtype situation the pre-rRNA folding state in early 35S containing LSU precursors might differ

significantly. I would suggest that the relevant statements are rephrased and corrected.

Results, paragraph entitled "Folding of individual domains in 25S": Please rephrase the first sentence "... proposed that the ordered assembly of ribosomal protein binding dictates".

Results, paragraph entitled "Folding of individual domains in 25S": In this section and the following one, Burlacu et al. compare timing of folding of individual domains with predictions derived from a model they refer to as "hierarchical r-protein assembly model". In the results section on page 11, on page 12 in the paragraph starting with "Collectively, all of the data..", in the abstract and in the discussion (p. 19-20) they refer then to this model and indicate possible inconsistency in regard to the dataset they obtained on pre-60s rRNA folding. The model referred to goes back to reference 32 (not to reference 21 which is among the citations in the beginning of the first paragraph on p.11) in which previous data on the role of ribosomal proteins for pre-rRNA processing, the binding sites of these proteins in ribosomes and some functional analyses on the hierarchy of ribosomal protein assembly events were summarized in a graphical abstract which can be interpreted in various ways. In reference 32 the authors also discuss a possible 5' - 3' gradient of 60S assembly. The graphical abstract in reference 32 and the term "5' - 3' assembly gradient" used in this manuscript might point to a highly coordinated sequence of events in which assembly in 5' regions of rRNA parts has to be completed before assembly (and possibly folding) of more 3' regions starts. Still, it also covers a scenario in which assembly of 5' regions has to progress to a certain degree before assembly (and possibly folding) of more 3' regions can be completed! The experimental results presented in reference 32 and in a number of other articles (published before and afterwards) analyzing the hierarchy of yeast 60S ribosomal protein assembly events do not at all support the first scenario. They rather point to the existence of several independent assembly trees in the various 60S rRNA domains (references 21, 22 and doi: 10.1371/journal.pone.0143768 and discussions therein). Hence, it seems therefore questionable that the authors of reference 32 intended this interpretation. Obviously, design and interpretation of graphical abstracts, including the ones in reference 32 and the ones shown in Burlacu et al. and in figures 1a and 8 can be misleading.

I have the impression that it would help the reader if the authors could differentiate more carefully between (1) the timing of progressive assembly of ribosomal proteins with pre-ribosomal populations in wildtype conditions (assembly timing) , (2) the role of ribosomal proteins in pre-rRNA processing, and (3) their impact on the stabilized pre-ribosomal association of other ribosomal proteins (assembly hierarchy). All these different aspects were studied in yeast in recent primary literature and they clearly relate to each other. It seems tempting, but as discussed below, is anyhow obviously misleading not to carefully distinguish between these aspects. To avoid any un-necessary confusion of the reader I would recommend not to use terms as "early r-proteins" or "early acting r-proteins" as in p. 7, p. 11, figure 1 and in other parts of the manuscript, without clearly defining whether this refers to assembly timing, involvement in processing steps, or a position in the assembly hierarchy. I will try to illustrate my point of view in the following: Hierarchical principles in 60s assembly are thought to apply to situations in which levels of an upstream ribosomal protein are limiting (with the term "upstream" designating the position in one of the trees of the 60S assembly hierarchy). That is in exponentially growing wildtype yeast analyzed by Burlacu et al. likely more the exception than the rule. Moreover, a rich source of primary data on the requirements of yeast (and mammalian) ribosomal proteins for 60S maturation indicated that, ultimately, misassembled particles are quite efficiently turned over and thereby eliminated from the steady state pool of pre-ribosomes analyzed in Burlacu et al. . The timing of assembly of yeast large ribosomal subunit proteins, which is indeed directly relevant for the interpretation of the data presented by Burlacu et al., was recently studied in some detail in reference 21 by a combination of biochemical approaches and proteomics. Very similar to the work presented here by Burlacu et al. the studies in reference 21 involved the comparison of pre-ribosomal populations of different maturation states isolated from wildtype yeast cells. It was concluded that "most of the LSU r-proteins start interacting with pre-ribosomes at early stages with

rather low affinity and that binding affinities increase with ongoing LSU maturation.." and "..thus these data indicated that most of the tested rpLs assemble with LSU precursors in a progressive way during their maturation". These interpretations about general characteristics of yeast 60S assembly are indeed well compatible with conclusions of Burlaku et al. on early 60S pre-rRNA folding. The results shown in ref 21 also clearly argued against the idea that requirement of a ribosomal protein for early, intermediate or late pre-rRNA processing steps allows a straight forward conclusion about respective early, intermediate or late assembly timing. The results disproved any kind of generalized "assembly on demand" principle for yeast 60s subunits. As example, yeast rpL1 was classified as very late acting ribosomal protein (doi: 10.1371/journal.pone.0008249, doi: 10.1371/journal.pone.0023579) but its general timing of ribosomal association parallels the one of most yeast 60S ribosomal proteins (including the group of 60S r-proteins required for early pre-ribosomal maturation) which start to associate with pre-ribosomes at early subunit maturation states (reference 21). In summary, I would recommend that the authors transfer the discussion of these aspects from the result to the discussion section which should allow them to develop these points more carefully.

Results, paragraph entitled "Formation of long-range interactions during early pre-60S assembly stages": In this part of the results section, in the figure legends section of figure 4 and in the abstract the authors interpret some of the changes in nucleotide conformational flexibility to be caused by "sequential binding" of individual ribosomal proteins to different 60S rRNA secondary structure domains. Indeed, progressive binding of individual r-proteins during 60S maturation was previously discussed to be related to a possibly stepwise establishment of interactions between such a ribosomal protein and different rRNA domains. The functional importance of some of these inter-domain bridges was previously analyzed in (truncation) mutants of yeast ribosomal proteins. The criteria applied by Burlacu et al. to trace this stepwise formation of inter-domain bridges in early 60S precursors are not very clear. They seem to focus on changes in nucleotide conformational flexibility at sites known to be contacted in mature ribosomes by ribosomal proteins. Did they just include sites of decreased nucleotide flexibility? Did they just include sites with proteinase k sensitive changes in nucleotide conformational flexibility? A crucial question in regard to their interpretation is how to assure that the observed changes in nucleotide conformational flexibility are indeed due to direct binding of the respective ribosomal proteins. The observed local changes in the RNA fold could be also caused by 1) protein independent RNA folding, 2) ribosomal protein binding at distal sites, and, importantly, 3) transient ribosome biogenesis factor binding at proximal or distal sites. That is obviously a matter of discussion and I wonder again whether this section of the manuscript would not suit much better to the discussion section. Option 3) is especially relevant for observed changes in nucleotide conformational flexibility of binding sites of rpL29, rpL40 and rpL42. As briefly discussed by the authors for rpL40, all three proteins were identified in references 21 and 34 and citations therein to belong together with a few others (including rpL10 and rpL24) to a small group of 60S ribosomal proteins with very specific assembly characteristics. They were observed to be substantially depleted from early pre-ribosomal populations analyzed by Burlacu et al. . Strikingly, in contrast to other 60S ribosomal proteins, this specific group of ribosomal proteins could also not be detected in high resolution cryo-EM structures of late nuclear 60S precursors (reference 13). As observed in late 60S precursors for rpL24, there binding sites might well be in early 60S precursor populations transiently occupied by ribosome biogenesis factors or, alternatively, remodelled in a factor dependent way. Results, paragraph entitled "Formation of long-range interactions during early pre-60S assembly stages": Correct phrasing of the sentence "Our data suggests that these r-proteins initially bind to a single domain (L17: domain I; L14, L33: domain VI) or in the 27SA2 pre-rRNA...".

Results, paragraph entitled "5.8S folding largely takes place during the conversion of 27SA2 to 27SB": It would clearly help in following this part of the result section if the authors could provide in figure 5a a larger, nucleotide resolved visualisation of the obtained data together with protein binding and secondary structure features. I miss here also the information on how the 5.8S folding state in (more)

mature(d) 60S subunits is reflected by the methods used by the authors to formally substantiate the conclusion that 5.8S rRNA folding largely takes place early in ribosome maturation. The statements about progressive establishment of r-protein:rRNA interactions are (see previous points for details) possibly again better suited for the discussion section.

Results, paragraph entitled "ITS2 and flanking regions fold into compact structures before the formation of 5.8S": My impression was that the word "formation" might be misunderstood in this context.

Results, paragraph entitled "ITS2 and flanking regions fold into compact structures before the formation of 5.8S": The authors introduce the term "ring-pin" model to refer to a secondary structure model of yeast ITS2 recently proposed by Coleman (reference 17). Later on, they refer to this model often as "Coleman model". Possibly it would be easier for the reader to go for one designation.

Results, paragraph entitled "ITS2 and flanking regions fold into compact structures before the formation of 5.8S": The sentence "In general, our ChemModSeq data (Fig. 6b) and follow-up primerextension data (Fig. 6c, Fig. S3a) agree best with the ring-pin structure proposed by Coleman17, which we also found to be consistent with previously published mutagenesis data38-40 (Fig. S3b)" refers to figures S3a and S3b, I guess that should be corrected.

Results, paragraph entitled "ITS2 and flanking regions fold into compact structures before the formation of 5.8S": I guess that the authors mean in the sentence "... and nucleotides that overlap L7 binding sites in region VI (Fig.S3), were more reactive" Rlp7 instead of L7.

Results, paragraph entitled "ITS2 and flanking regions fold into compact structures before the formation of 5.8S": The authors say that "It was previously proposed that ITS2 undergoes a conformational change from an open to a more helical structure prior to cleavage of the 27SB pre-rRNA at site C2" and cite references 36-38. I am not sure that in references 36 and 37 such a transition was proposed.

Results, paragraph entitled "ITS2 and flanking regions fold into compact structures before the formation of 5.8S": The authors state that "Dissociation of these factors is required for subsequent removal of ITS2 after C2 cleavage" and cite reference 13 to substantiate this statement. I guess that relates to rather extended binding interfaces between some factors and the 5.8S and 25S proximal ITS2 stems. I am not sure that there is experimental evidence clarifying the relationship between factor dissociation and initial removal of other ITS2 parts.

Results, paragraph entitled "ITS2 and flanking regions fold into compact structures before the formation of 5.8S": The authors conclude here that "ITS2 is generally highly structured and does not appear to undergo any major changes in secondary structure during posttranscriptional conversion of 35S to 27SB". When looking at Figure 6b and when comparing the patterns of ITS2 SHAPE reactivities in the first two lanes (35S and 27SA2) with the 3rd lane (27SB) there are significant differences apparent. If I am not getting it wrong, in the 27SB sample there are often (especially in the region spanning nucleotide 100 to 200) regions of strongly increased nucleotide flexibility which are partially proteinase k sensitive. Were these changes re-analyzed by primer extension analyses? I think the authors point to this observation in the sentence "This again suggests that ITS2 may become slightly more flexible prior to formation of 27SB". Can we exclude that this reflects significant changes in ITS2 secondary structure which are induced by proximal or distal binding of proteins?

Discussion, paragraph entitled "Folding of 5.8S largely takes place during the conversion of 27SA2 to 27SB": At the end of this paragraph and in figure 8 the authors develop a model which can be

understood in the sense that stable assembly of rpL17, rpL35 and rpL37 paves the way for the assembly of late B-factors and subsequently leads to processing of the 3' ITS1 and formation of 27SB (last part of this section of the discussion). When looking at the primary literature I had the impression that ITS1 processing can go on rather well in yeast expression mutants of rpL17, rpL35, rpL37 or "late B-factors". In some of them even substantial cleavage at C2 was observed. How does that relate to the model?

Discussion, paragraph entitled "The temporal formation of (long-range) r-protein-rRNA interactions " : That paragraph recapitulates some of the statements made in the respective two parts of the results section. Specific comments on the relevant topics were given further above.

Reviewer #2 (Remarks to the Author):

Eukaryotic ribosome biogenesis is an extremely complicated process. Most trans-acting factors involved have been identified. The first high-resolution structures of pre-ribosomes have been determined, providing initial insights into the function of several assembly factors.

Here the authors have addressed the dynamics of some of the restructuring events that occur during subunit assembly. They have applied chemical probing (SHAPE) on three isolated heterogeneous populations of pre-ribosomes, each amino-purified from a specific bait (Mrd1, Rrp5, and Nsa2).

There is a lot of work involved, and it is generally of good quality.

My main problem is that the authors are making numerous (too many in my opinion) assumptions to interpret their data (see pages 8 and 9). The description of the different categories of hits (classification) is also exceedingly complicated. The authors are making a few predictions based on their data but, regretfully, they do not validate any of them.

In a subsection of their work, the authors are challenging a published model for ITS2 processing (so-called ring to hairpin model). Challenging a published model is of course completely fine, although in this case it is quite well-supported, and it has been used extensively by the authors themselves in the past. The authors should discuss whether their revised ITS2 folding model impact (?or not) their former conclusions (see Granneman et al EMBO J. 2011).

In conclusion, the manuscript suffers from the highly speculative nature of the conclusions, it is very difficult to read, and altogether, the biological significance of the work is limited.

Comments on the methodology

-There is an important inherent limitation to the method: purifying pre-ribosomal particles from one assembly factor, and showing the RNA to be intact (by mapping its 5'-end by primer extension, and showing its size on a bioanalyzer chip) does not mean that a single type of pre-ribosome was purified. The authors are applying a single nucleotide resolution chemical mapping technique to heterogeneous populations of particles (they are doing population analysis not single particle analysis) that have, by definition, undergone various levels of restructuring events. This should be discussed.

-Is it a problem that the Mrd1 bait is in fact a mutated construct (Mrd1 RBD5delete)?

-A new nomenclature for ribosomal proteins has been proposed based on extensive structural work and phylogeny, using it means making our field less obscure.

Reviewer #3 (Remarks to the Author):

Here Burlacu et al have used RNA structure probing to study early RNA folding in nucleolar pre-60S particles. In particular the authors have focused on the structural changes around ITS2 and the 5.8S rRNA. The data presented in this manuscript suggest that the 5.8S rRNA adopts its structure predominantly during the 27SA2 to 27SB transition while ITS2 adopts a compact structure early on. Overall this manuscript contains important new insights that will be very important for the field and it is suitable for publication in Nature Communications provided that the following points are addressed by the authors in a revised version.

1. In Figures 1 and 2 the authors argue that the selection of Mrd1, Rrp5 and Nsa2 will result in the isolation of 35S, 27SA2 and 27SB particles only but insufficient evidence is presented to show this. Primer extensions are used only for particles isolated via tagged Rrp5 and Nsa2 to show that the associated pre-rRNAs terminate at sites A2 and BL/BS respectively. Comparative mass spectrometry is used to compare A3 and B factors only between Rrp5 and Nsa2.

First, the authors should provide evidence (northern blotting etc.) to show that the isolated Mrd1-associated particle indeed contains a 35S pre-rRNA. In addition the authors should show the Mrd1-associated treated pre-rRNA in Figure 2a next to the Nsa2 and Rrp5 associated pre-rRNA species.

Second, the authors should provide SDS-PAGE analysis of all three isolated particles to address the homogeneity of each isolated particle. This will be important to ensure that a homogenous population of particles has been isolated for subsequent RNA analysis. This analysis will further address the following issue:

Rrp5 is associated with early ribosome assembly intermediates including both early pre-40S particles (SSU processomes) as well as early pre-60S particles. By purifying Rrp5-associated particles, the authors clearly co-purify SSU processome components (as shown in the supplement), which presumably contain a pre-rRNA containing the 5'ETS and 18S rRNA. Alternatively this sample may contain a 35S pre-rRNA species.

The authors should provide further evidence of which other pre-rRNA is present besides 27SA2.

2. In panel 1c and 1d it is unclear which data point refers to which ribosome assembly factor and labels should be added.

3. The positions of bases with changed flexibility are very hard to follow in the three panels of Figure 3c.

This figure could be improved by using more traditional views (such as crown view etc.). Mapping the indicated bases on an associated secondary structure diagram, such as the one shown in Figure 3a, would be very helpful to correlate the position of the bases from 2D to 3D.

4. In the current manuscript the authors have not adequately credited a very recent cryo-EM structure of a late nuclear pre-60S particle (briefly mentioned as reference 13). This relates in particular to the (secondary) structure of ITS2. Importantly, the cryo-EM structure is in very nice agreement with the RNA structure probing data presented in the current manuscript and only agrees with the ring-pin model shown in Figure 6a. Here it would be appropriate to show the 3D structural model of ITS2 and the associated ribosome assembly factors (pdb code: 3JCT) with the RNA structure probing data.

5. Lastly, there are some minor corrections that need to be addressed, such as the following typos.

- a. On page 11: "the the observed increase"
- b. On page 13: "(Fig. 5d, purple nucleotides)" - the nucleotides are green
- c. On page 16: "our data suggests that L26 contact"

Reviewer #4 (Remarks to the Author):

Review of Burlacu et al.

This is a highly interesting paper from the Granneman Lab presenting an analysis of 60S assembly based on affinity purification of the ribosomal complexes and RNA structure probing.

The method employed is similar to the method that Granneman and coworkers previously used to analyse the 40S subunit (Hector et al., 2014). The major finding of the study is that folding of the 5.8S rRNA occurs at a specific stage of processing, suggesting that subsequent nuclear processing events depend on the 5.8S folding to be completed. Moreover, the study provides new information about the timing of the formation of protein-RNA interactions and long range RNA-RNA interactions. These findings are clearly novel and will be of high interest in the specific field. However, in this reviewer's opinion there are some methodological issues that will have to be addressed before this paper can be published.

The purification of the ribosomal complexes is quite convincing and form a strong basis for the following probing experiments. While the methods used for the proteomics experiment are described in details, the description of the methods used for analysis of the SHAPE data is very short and makes it very difficult for the reader to evaluate the potential for bias in the data and indeed to reproduce the work presented in the paper. For the calculation of SHAPE reactivities the authors cite Deigan et al. 2009 and Tang et al. 2015, however, although these papers are conceptually quite similar, the described strategies also contain differences and user specified thresholds that could potentially influence the reactivity data. A detailed description of the data analysis that is performed in this paper is therefore absolutely essential before this paper is published.

Based on the Deigan and Tang references, the analysis used in the manuscript must be based on a log ratio between normalized counts in the probed sample and the control. This is in contrast to the strategy that were used in the previous paper (Hector et al). The log ratio approach is more susceptible to biases introduced by differences in the coverage between the control and probed sample. Such coverage differences could arise from RNA degradation or from the probing itself. The latter is a real concern as the 1M7 reagent is used in a high concentration in this study (30mM final). If coverage differences are present between control and sample and vary between the ribosomal complexes, this could lead to specific regions having lower or higher reactivities. This is a concern with respect to the 5.8S, where this kind of general trend is observed by the authors. As the change in 5.8S flexibility during assembly is a major claim of the paper, the authors need to rule out that differences in coverage of the probing data affects their data.

The obtained sequencing data is paired-end and it is therefore surprising that the authors are not using the normalization methodology that were developed in the Hector et al study, which should be more robust with respect to coverage and in addition uses a nice Poisson expectation maximization algorithm for scoring nucleotides modified or non-modified. I suggest that this method is applied for the data analysis.

In this study, the SHAPE reactivities of the different samples are used to classify positions into modified vs non-modified based on an arbitrary cut-off. How the cut-off is chosen is not discussed, but

should be clarified as it could influence the resulting data a lot. The actual cut-off used is a bit unclear from the paper, which states: "The nucleotides with 1M7 reactivities below the first quartile were considered unmodified. Nucleotides with 1M7 reactivities above the first quartile were considered modified, while the nucleotides with 1M7 reactivities above the third quartile were considered highly modified.". In the main text it is stated that 1940 nucleotides were found to react with 1M7, but in the Table S1, samples have 1294 to 1405 out of the 3786 nucleotides that are called modified. Neither of these numbers seem to be close to either $\frac{1}{4}$ of the data points or $\frac{3}{4}$ of the data points, which should be explained. Again the previously developed method using Poisson expectation maximization should be superior to the one use here.

The authors do not attempt to validate the quality of their structure probing data. As a minimum, it would be relevant to test if the authors reactivity data significantly correlate with the crystal structure basepair annotation and when the authors identify sets of nucleotides that seem to change their as a result of protein binding, it would be relevant to demonstrate that this set is significantly closer in space to protein in the relevant ribosomal complexes than other positions.

In Hector et al. in vivo probing with NAI is performed and inclusion of this kind of data for the non-proteinase K samples would certainly improve the paper, as it is unclear how purification affects the RNA structure of the different ribosomal complexes.

Minor points.

In figure 3b, low coverage should be removed, because otherwise the figure is very difficult to interpret.

The yeast rRNA crystal structure paper should be cite in figure texts.

The method used here so different from Hector et al. that is does not makes sense to use the ChemModSeq name about the experiments.

Figure 2b is not especially intuitive and should be modified.

Figure 1c: I am not completely convinced that it is a good idea to compare spectrum counts for different proteins. Would it make sense to compare the spectrum count for the same protein in the two pull downs and then subsequently evaluate if they are A3 or B?

The title is "High-throughput RNA structure probing reveals critical folding events during early stages of ribosome biogenesis in yeast". Here, I feel that critical should be removed as the authors not really demonstrate the folding is a critical event.

Reviewers' comments:

Reviewer #1 (Remarks to the Author):

1. Burlaco et al. analyze in the submitted manuscript changes in the folding states of rRNA precursors during early steps of large ribosomal subunit maturation in the yeast *S. cerevisiae*. They affinity purified from yeast cells three different early to intermediate large ribosomal subunit precursor populations and probed them in vitro with the agent IM7 to determine changes in conformational flexibility of individual pre-rRNA nucleotides. The obtained dataset represents an important resource for people interested in early yeast large ribosomal subunit maturation and provides new insights into this process!

We are very grateful for the constructive criticism this reviewer has provided as well as the thorough review of our manuscript!!!

2. Specific comments and questions (in order of appearance of relevant issues in the manuscript):
Introduction, p.4: "To generate an overview pre-rRNA folding steps during the early stages of 60S maturation on a single nucleotide resolution level ...".
Phrasing should be corrected.

We have rewritten large parts of the introduction and have corrected the phrasing on page 3 of the Introduction.

3. Results: In general, my impression was that the manuscript would benefit from largely shortening the results section and focussing there on a neutral description of experimental strategies applied and the results obtained. Other parts of the results section (see below), which mainly provide possible interpretation of the data and of relevant literature, could be extended and moved to the discussion section.

We have now moved large sections of the results where we describe our findings to the Discussion where we also now discuss alternative interpretations of the data and highlight other relevant literature.

4. Results, paragraph entitled "Purification and characterization of pre-60S intermediates for RNA structure probing": The authors use a *mrd1-dRBD5* mutant strain in which 35S pre-rRNA processing is delayed to facilitate efficient purification of early 35S containing pre-ribosomes. They compare conformational flexibility of rRNA nucleotides in the resulting particles with the ones in more matured large subunit precursors. It is unclear how much of the detected changes can be attributed to the mutant situation and how much reflect changes in pre-rRNA folding happening during the 35S to 27SA transition in a wildtype situation. This point should be communicated to the reader to avoid misinterpretation of the presented dataset.

We completely agree with Reviewer 1 that the folding kinetics could be affected by the Mrd1 mutation, but based on the following we believe that it is unlikely to have a major effect: Mrd1 is associated with precursors containing the 18S rRNA but is not required for synthesis of 25S rRNA. The Mrd1 mutant used here (Mrd1 Δ 5) accumulates modest levels of 35S but does not significantly affect the levels of 27S and 7S pre-rRNAs as well as the 25S rRNAs (Lundkvist et al 2009). It is therefore reasonable to assume that the rRNA folding steps in the 27S pre-rRNA are (largely) unaffected. However, we agree with Reviewer 1 that we cannot completely exclude the possibility that the mutation will affect the kinetics of LSU rRNA folding. To try to resolve this issue, we tried to further optimize the immunoprecipitation steps to see if we could recover native 35S from TAP-tagged strains. FL managed to purify a small quantity of 35S from Rrp5 immunoprecipitations, which was sufficient for performing primer extension reactions but not for library preparations. FL compared the 35S isolated from the Rrp5-TAP IP with 35S from the Mrd1 Δ 5 strain and the results show that in the 5.8S region we did not see any noticeable differences between the samples. We therefore conclude that it is unlikely that the Mrd1 Δ 5 mutation has a significant effect on the folding kinetics of the 35S and 27S pre-rRNA species. These new findings have been included in the discussion of the manuscript and the 5.8S primer extension results are presented in the new Figure S7.

5. Results, paragraph entitled "Purification and characterization of pre-60S intermediates for RNA structure probing": Is there a reason that the authors characterized the RNA and protein content of Rrp5 and Nsa2 particles but not of mrd1-dRBD5 particles (see Figure 1b-d)?

The main reason why we only focused on the Rrp5 and Nsa2 particles because previously we had done mass-spectrometry on Mrd1 particles (Segerstolpe et al NAR 2013) and found that pre-60S assembly factors were highly underrepresented, making a direct comparison between this particle and the other pre-60S particles very challenging. Secondly, most of the rRNA structural changes occurred during the conversion of 27SA₂ to 27SB.

6. Results, paragraph entitled "Purification and characterization of pre-60S intermediates for RNA structure probing": In Fig. 1e the result of the analysis of the ribosomal protein content of Fun12 associated ribosomal particles is shown. The reason for performing this experiment is not explained in the text and I could also not find any other reference to this experiment in the results or discussion sections. I guess the Fun12 particles should reflect (more) mature(d) 60S subunits. In this regard, why do they seem depleted of several ribosomal proteins which were detected in precursor particles (L9, L43, L38, L35, L34 and others)?

Yes, this is correct. Fun12 was used to isolate 60S translation initiation complexes, which were used as a source for mature 60S subunits. We have now clarified this in the text on page 6 of the revised manuscript. The peptides

generated by the ribosomal proteins are already not very abundant in the Rrp5 and Nsa2 particles, which are composed of 425 and 170 proteins respectively; the Fun12 particle is composed of 683 identified proteins. Moreover, unlike Rrp5 and Nsa2, Fun12 is found in an associated and unassociated form, so the vast majority of peptides identified in this run were from Fun12. Therefore, the peptides from the R-proteins (L9, L43, L38, L35, L34 and others) that were not identified in the Fun12 particle were likely missed by the Q-exactive due to dynamic range detection problems. Peptide separation using MudPIT could have improved the detection of ribosomal proteins in the purifications but required instruments were not accessible.

7. Results, paragraph entitled "Purification and characterization of pre-60S intermediates for RNA structure probing": Is there a reason that different buffers were used for affinity purification of particles for proteome analysis (buffer RNP150 and buffer TBT150) when compared to buffers used for purification of particles used for RNA structure probing experiments (buffer TMN150)?

The Oeffinger group has tried many different buffer compositions (including the TMN150) and found that for the pre-60S complexes the buffer composition (within reasonable limits) does not make any difference. While there is a difference in $MgCl_2$ concentration in the buffers used, the overall buffering capacity, salt and detergent concentrations are the same or similar. They have not seen a significant difference in complex composition for these complexes with $MgCl_2$ concentrations below 7.5mM. They have done a large number of purifications in parallel and decided to use the same buffer for all pre-60S particles they have analyzed. This is why they did not use TMN150.

8. Results, paragraph entitled "Purification and characterization of pre-60S intermediates for RNA structure probing": The authors did not include in their analysis large subunit precursors downstream of 27SB pre-rRNA. This might have further supported their notion that significant aspects of large ribosomal subunit pre-rRNA folding occur at early stages of maturation. It could have also served as further control for their approach and might have substantiated several aspects of the interpretation of their results. The authors might explain to the readers their choice to focus on the analysis of early 60S precursors.

We completely agree with Reviewer 1. Although our work generated very interesting results, it was troubled by many technical challenges, some of which we were not able to resolve. EB and FL spent a long time optimizing the purification conditions for probing these particles and found that not all particles were amendable for structure probing analyses. The reason why we could not analyze later pre-60S particles is because for the chemical probing experiments we had to gel-purify the pre-rRNA species of interest to remove contaminating mature ribosomal RNAs (25S and 5.8S). Unfortunately, the later pre-60S particles contain 5' extended 25S species that cannot be separated from the mature 25S by agarose gel electrophoresis. We tried to analyze pre-rRNAs

purified after a two-step purification, but this did not significantly improve the purity and gave us poor quality RNAs for structure probing. In the end we decided therefore to focus on the nucleolar particles, which contain much longer pre-rRNA species. This has now been clarified in the Discussion of the revised manuscript in the paragraph "Purification of ribosome assembly intermediates".

9. Results, paragraph entitled "ChemModSeq analyses of purified pre-60S ribosomes": The authors probe nucleotide flexibility in ex vivo isolated pre-ribosomal populations treated or not with proteinase K. They use differences in nucleotide flexibility between these two conditions to distinguish "rRNA folding events" and "protein binding events" (see for example Fig. 2b). Proteinase K treatment was done, according to the reference cited, for 3 hours at 14°C. Did the authors test whether protein digest in these conditions was complete or whether partially protected parts of proteins remained associated with rRNA with possible influence on its structure?

Yes, we analyzed the isolated proteins before and after Proteinase K treatment by SDS-PAGE, which showed that the enzymes very effectively degraded the proteins from the affinity purified material. This result is now shown in the new Supplementary Figure S1f. We agree with Reviewer 1 that we cannot completely exclude the possibility that short peptides remain associated with the pre-rRNA and that this could influence the structure probing results. However, considering the conditions in which the digestion was performed (1% SDS, 150 mM NaCl), we feel that it is reasonable to assume that the vast majority of the protein-RNA interactions has been disrupted. We now also use throughout the manuscript the more neutral PK-sensitive and PK-insensitive terms when comparing the Proteinase-K data with the results from the whole particles (also see the next comment). We hope that this will alleviate Reviewer 1's concern as in this way we are not claiming that all peptides have dissociated.

10. The authors classify "rRNA folding events" as the ones for which changes in nucleotide flexibility from earlier to later pre-ribosomal populations was observed which resisted proteinase K treatment. That criterion seems quite disputable. Beside technical issues, as partial proteinase K digest (see above), the authors themselves discuss in this paragraph scenarios leading to false classification, as transient protein-RNA interactions with lasting impact on RNA fold. I would much prefer neutral terms to describe these classes, as "proteinase k sensitive" and "proteinase k insensitive" together with an appropriate discussion of their possible interpretation in the discussion section. That could also help to shorten this quite lengthy part of the results section. One could briefly introduce the proteinase k experiment as an approach to detect protein dependent rRNA folding states and come back to it in the discussion section. The following section listing all possible combinatorial categories together with some (but not all) possible interpretations could then be drastically shortened.

This is an excellent suggestion. We have now substantially shortened the results

section and made changes to Figure 2. On pages 7 and 8 we now only refer unchanged, proteinase K sensitive and proteinase K insensitive scenarios and have included a more detailed explanation of these results in the Discussion section. We hope that this makes the interpretation of our results easier to follow.

11. Results, paragraph entitled "ChemModSeq analyses of purified pre-60S ribosomes": The following sentence contains a duplication: "The fact that the 1M7 reactivity does not (dramatically) change when deproteinizing the samples, however, suggests that the formed structure does not require the presence of the proteins to remain stable, suggested by the fact that the 1M7 reactivity does not (dramatically) change when deproteinizing the samples."

See previous answer. The section that included this sentence has now been removed.

12. Results, paragraph entitled "ChemModSeq analyses of purified pre-60S ribosomes": Supplementary table S2, containing 1M7 reactivities per nucleotide, would benefit from a more detailed legend. That is important if the dataset should serve as a resource for the community.

We have simplified the headers in the table and have provided a more detailed description of each column in the Supplementary Table 2 legend. Furthermore, we also included a table in the "Categorizations" sheet that explains in detail how each nucleotide was classified. We hope that this information will make it easy for the reader to interpret the results.

13. Results, paragraph entitled "ChemModSeq analyses of purified pre-60S ribosomes": The figure legend of figure 2 a) needs rephrasing.

We have made a number of changes to the legend of Fig. 2 to make it more clear. We had also accidentally omitted the legend for Fig. 2c, which has now been included.

14. Results, paragraph entitled "ChemModSeq analyses of purified pre-60S ribosomes": "strikingly, 40% of the analyzed nucleotides .." and Introduction p.4: "Our analyses suggest that roughly one third of 60S pre-rRNA folding occurs cotranscriptionally..". In the results part the authors argue that 40% of all the analysed nucleotides did not show any changes in 1M7 reactivity in the pre-rRNA populations analyzed and, in their opinion, that suggest that a significant fraction of the 27S pre-rRNA already folds co-transcriptionally (results page 10). Do we know the amount of changes in conformational flexibility happening in pre-ribosomal populations downstream of the ones analyzed here? And how do the authors quantify changes in "60S pre-rRNA folding" states which include changes in the tertiary structure exemplified by drastic rotations of parts of the central protuberance observed at later stages of 60S maturation?

These are valid points. Indeed, we do not know how many changes in nucleotide flexibility occur in later particles and therefore it is possible that some nucleotides that did not show any changes in early particles might still undergo conformational changes in later particles. Unfortunately, as mentioned in point 8, due to technical reasons we were unable to probe later assembly intermediates. It is not known how rotations in 3D affect the flexibility of nucleotides, however we predict that most of the changes would be found at the base where the rotation takes place. We therefore removed the statements that a large fraction of 27S folds co-transcriptionally from the introduction and the results section.

15. The authors use moreover a mutant in which pre-rRNA processing is delayed to be able to purify efficiently the earliest (posttranscriptional) LSU precursor included in their analysis (see also discussion below). This delay in pre-rRNA processing obviously might increase the time for pre-rRNA to fold in early pre-ribosomal populations. Therefore, and due to various other reasons in a wildtype situation the pre-rRNA folding state in early 35S containing LSU precursors might differ significantly. I would suggest that the relevant statements are rephrased and corrected.

See comment number 4

16. Results, paragraph entitled "Folding of individual domains in 25S": Please rephrase the first sentence " ... proposed that the ordered assembly of ribosomal protein binding dictates".

This has been changed in the text.

17. Results, paragraph entitled "Folding of individual domains in 25S": In this section and the following one, Burlacu et al. compare timing of folding of individual domains with predictions derived from a model they refer to as "hierarchical r-protein assembly model". In the results section on page 11, on page 12 in the paragraph starting with "Collectively, all of the data..", in the abstract and in the discussion (p. 19-20) they refer then to this model and indicate possible inconsistency in regard to the dataset they obtained on pre-60s rRNA folding. The model referred to goes back to reference 32 (not to reference 21 which is among the citations in the beginning of the first paragraph on p.11) in which previous data on the role of ribosomal proteins for pre-rRNA processing, the binding sites of these proteins in ribosomes and some functional analyses on the hierarchy of ribosomal protein assembly events were summarized in a graphical abstract which can be interpreted in various ways. In reference 32 the authors also discuss a possible 5'- 3' gradient of 60S assembly. The graphical abstract in reference 32 and the term "5' - 3' assembly gradient" used in this manuscript might point to a highly coordinated sequence of events in which assembly in 5' regions of rRNA parts has to be completed before assembly (and possibly folding) of more 3' regions starts. Still, it also covers a scenario in which assembly of 5' regions has to progress to a certain degree before assembly (and

possibly folding) of more 3' regions can be completed! The experimental results presented in reference 32 and in a number of other articles (published before and afterwards) analyzing the hierarchy of yeast 60S ribosomal protein assembly events do not at all support the first scenario. They rather point to the existence of several independent assembly trees in the various 60S rRNA domains (references 21, 22 and doi: 10.1371/journal.pone.0143768 and discussions therein). Hence, it seems therefore questionable that the authors of reference 32 intended this interpretation. Obviously, design and interpretation of graphical abstracts, including the ones in reference 32 and the ones shown in Burlacu et al. and in figures 1a and 8 can be misleading. I have the impression that it would help the reader if the authors could differentiate more carefully between (1) the timing of progressive assembly of ribosomal proteins with pre-ribosomal populations in wildtype conditions (assembly timing), (2) the role of ribosomal proteins in pre-rRNA processing, and (3) their impact on the stabilized pre-ribosomal association of other ribosomal proteins (assembly hierarchy). All these different aspects were studied in yeast in recent primary literature and they clearly relate to each other. It seems tempting, but as discussed below, is anyhow obviously misleading not to carefully distinguish between these aspects. To avoid any un-necessary confusion of the reader I would recommend not to use terms as "early r-proteins" or "early acting r-proteins" as in p. 7, p. 11, figure 1 and in other parts of the manuscript, without clearly defining whether this refers to assembly timing, involvement in processing steps, or a position in the assembly hierarchy. I will try to illustrate my point of view in the following: Hierarchical principles in 60s assembly are thought to apply to situations in which levels of an upstream ribosomal protein are limiting (with the term "upstream" designating the position in one of the trees of the 60S assembly hierarchy). That is in exponentially growing wildtype yeast analyzed by Burlacu et al. likely more the exception than the rule. Moreover, a rich source of primary data on the requirements of yeast (and mammalian) ribosomal proteins for 60S maturation indicated that, ultimately, misassembled particles are quite efficiently turned over and thereby eliminated from the steady state pool of pre-ribosomes analyzed in Burlacu et al. The timing of assembly of yeast large ribosomal subunit proteins, which is indeed directly relevant for the interpretation of the data presented by Burlacu et al., was recently studied in some detail in reference 21 by a combination of biochemical approaches and proteomics. Very similar to the work presented here by Burlacu et al. the studies in reference 21 involved the comparison of pre-ribosomal populations of different maturation states isolated from wildtype yeast cells. It was concluded that "most of the LSU r-proteins start interacting with pre-ribosomes at early stages with rather low affinity and that binding affinities increase with ongoing LSU maturation.." and "..thus these data indicated that most of the tested rpLs assemble with LSU precursors in a progressive way during their maturation". These interpretations about general characteristics of yeast 60S assembly are indeed well compatible with conclusions of Burlaku et al. on early 60S pre-rRNA folding. The results shown in ref 21 also clearly argued against the idea that requirement of a ribosomal protein for early, intermediate or late pre-rRNA processing steps allows a straight

forward conclusion about respective early, intermediate or late assembly timing. The results disproved any kind of generalized "assembly on demand" principle for yeast 60s subunits. As example, yeast rpl1 was classified as very late acting ribosomal protein (doi:10.1371/journal.pone.0008249, doi:10.1371/journal.pone.0023579) but its general timing of ribosomal association parallels the one of most yeast 60S ribosomal proteins (including the group of 60S r-proteins required for early pre-ribosomal maturation) which start to associate with pre-ribosomes at early subunit maturation states (reference 21). In summary, I would recommend that the authors transfer the discussion of these aspects from the result to the discussion section which should allow them to develop these points more carefully.

We thank the reviewer for pointing this out. We certainly do not want to create confusion and we agree that we should have more carefully defined what we meant with early, middle and late acting r-proteins. To address this reviewer's concern (and also to simplify the story) we have removed all references to early-middle-late acting/assembling r-protein in both figures and text. This reviewer is correct that all of the available data suggests that most r-proteins assembly together quite early and that their binding gradually stabilizes during the assembly.

We completely agree with the model put forward by this reviewer that it is unlikely that 60S assembly follows an "assembly on demand" principle. However, we do feel that it is important to discuss the hierarchical model put forward by Woolford and colleagues as they clearly claim that assembly occurs in such hierarchical fashion (which could be interpreted as "assembly on demand") and that this is coupled to the stages in which these r-proteins function in pre-rRNA processing. This has also recently been discussed in detail in a recent review by de la Cruz and colleagues. On page 115 in this review they state: "*These results suggest that assembly of LSUs may proceed in a hierarchical fashion, beginning with the solvent-exposed surface, followed by the PET, the intersubunit interface, and finally the CP*".

As suggested by Reviewer 1, we have now moved all the debate related to these aspects to the Discussion section. We have also made considerable changes to the main text and Figures to address the points raised by this reviewer. On page 8 and 9 of the revised manuscript, we have renamed the title of the paragraph "*Folding of individual domains in 25S*" into "*Gradual assembly of r-proteins into 60S pre-ribosomes*". In this paragraph we now discuss how the ChemModSeq data helped us to make predictions about r-protein-rRNA interactions, by focusing on the proteinase K sensitive sites that overlap with known r-protein interaction sites.

18. Results, paragraph entitled "Formation of long-range interactions during early pre-60S assembly stages": In this part of the results section, in the figure legends section of figure 4 and in the abstract the authors interpret some of the changes in nucleotide conformational flexibility to be caused by "sequential binding" of individual ribosomal proteins to different 60S rRNA secondary structure domains.

Indeed, progressive binding of individual r-proteins during 60S maturation was previously discussed to be related to a possibly stepwise establishment of interactions between such a ribosomal protein and different rRNA domains. The functional importance of some of these inter-domain bridges was previously analyzed in (truncation) mutants of yeast ribosomal proteins. The criteria applied by Burlacu et al. to trace this stepwise formation of inter-domain bridges in early 60S precursors are not very clear. They seem to focus on changes in nucleotide conformational flexibility at sites known to be contacted in mature ribosomes by ribosomal proteins. Did they just include sites of decreased nucleotide flexibility? Did they just include sites with proteinase k sensitive changes in nucleotide conformational flexibility?

We assumed that only changes in reactivity that were *sensitive* to proteinase K digestion could be protein-dependent changes in rRNA flexibility. These sites also had to overlap with binding sites for known ribosomal proteins. We focused our analyses on positions where we could see a decrease in flexibility in the particles but this flexibility was restored in the proteinase K treated samples. We also looked for when the changes occurred in the nucleotides that belonged to different structural domains. In some cases, we could also see an increase in nucleotide flexibility that was sensitive to proteinase K treatment and many of these nucleotides are bound to r-proteins in the crystal structure. These may represent cases where protein binding stabilizes a nucleotide conformation that is favorable to 1M7 modification.

We hope this clarifies this issue.

19. A crucial question in regard to their interpretation is how to assure that the observed changes in nucleotide conformational flexibility are indeed due to direct binding of the respective ribosomal proteins. The observed local changes in the RNA fold could be also caused by 1) protein independent RNA folding, 2) ribosomal protein binding at distal sites, and, importantly, 3) transient ribosome biogenesis factor binding at proximal or distal sites. That is obviously a matter of discussion and I wonder again whether this section of the manuscript would not suit much better to the discussion section. Option 3) is especially relevant for observed changes in nucleotide conformational flexibility of binding sites of rpL29, rpL40 and rpL42. As briefly discussed by the authors for rpL40, all three proteins were identified in references 21 and 34 and citations therein to belong together with a few others (including rpL10 and rpL24) to a small group of 60S ribosomal proteins with very specific assembly characteristics. They were observed to be substantially depleted from early pre-ribosomal populations analyzed by Burlacu et al. Strikingly, in contrast to other 60S ribosomal proteins, this specific group of ribosomal proteins could also not be detected in high resolution cryo-EM structures of late nuclear 60S precursors (reference 13). As observed in late 60S precursors for rpL24, their binding sites might well be in early 60S precursor populations transiently occupied by ribosome biogenesis factors or, alternatively, remodelled in a factor dependent way.

These are excellent points. We now address this in detail on page 17 and 18 in the Discussion section of the revised version of the manuscript.

20. Results, paragraph entitled "Formation of long-range interactions during early pre-60S assembly stages": Correct phrasing of the sentence "Our data suggests that these r-proteins initially bind to a single domain (L17: domain I; L14, L33: domain VI) or in the 27SA2 pre-rRNA...".

We thank this reviewer for pointing this out. The sentence has been rephrased on page 9 of the revised version.

21. Results, paragraph entitled "5.8S folding largely takes place during the conversion of 27SA2 to 27SB": It would clearly help in following this part of the result section if the authors could provide in figure 5a a larger, nucleotide resolved visualisation of the obtained data together with protein binding and secondary structure features.

We made a larger figure as this reviewer requested, however, adding the individual nucleotide positions, secondary structure annotation and r-protein binding sites made the figure very busy and the individual nucleotide numbering was difficult to read. So we now increased the size of the figure as much as possible but left out the individual nucleotide numbering. If the reader wants to look at individual nucleotide positions, we would recommend them to look Supplementary Table 2, which has all this information as well as the nucleotide sequence in it. In the revised manuscript, we not only increased the size of the heat map in Figure 5a but we also added an extra column with secondary structure information (last column in the heat map). White blocks are regions that, according to the secondary structure model, are single stranded. Blue blocks are the double-stranded regions.

22. I miss here also the information on how the 5.8S folding state in (more) mature(d) 60S subunits is reflected by the methods used by the authors to formally substantiate the conclusion that 5.8S rRNA folding largely takes place early in ribosome maturation.

As stated in point 8, we were unable to perform structure probing experiments on later LSU assembly intermediates so we have softened our statements in this paragraph. We now state on page 10 of the revised manuscript: *"The ChemModSeq and subsequent follow-up primer extension data suggest that 5.8S region undergoes considerable structural rearrangements during the conversion of 35S to 27SB pre-rRNA, with major changes occurring specifically during the conversion of 27SA₂ to 27SB"*.

Recently the Woolford and Gao labs published a high-resolution structure of a Nog2 pre-60S particle that has a similar protein composition as the Nsa2 particles we analyzed. By comparing the 5.8S structure from the EM data with

the structure in the ribosome, we found that both structures were remarkably similar (see new Fig. S4). Therefore, we feel that it is reasonable to conclude that most of the restructuring in 5.8S takes place during the conversion of 27SA₂ to 27SB. We now briefly discuss this on page 11 of the revised manuscript.

23. The statements about progressive establishment of r-protein:rRNA interactions are (see previous points for details) possibly again better suited for the discussion section.

We agree that much of the conclusions drawn in the results section could be moved to the discussion section. However, in this case we believe that moving the discussion about the r-proteins binding to 5.8S would essentially mean that we would not have much to conclude about figure 5f. In this case, we would prefer to keep the text in this paragraph and make a general statement about gradual r-protein assembly in the discussion.

24. Results, paragraph entitled "ITS2 and flanking regions fold into compact structures before the formation of 5.8S": My impression was that the word "formation" might be misunderstood in this context.

We agree. We have now changed the title of the paragraph to: *"ITS2 and flanking regions form compact structures before major restructuring of 5.8S"*.

25. Results, paragraph entitled "ITS2 and flanking regions fold into compact structures before the formation of 5.8S": The authors introduce the term "ring-pin" model to refer to a secondary structure model of yeast ITS2 recently proposed by Coleman (reference 17). Later on, they refer to this model often as "Coleman model". Possibly it would be easier for the reader to go for one designation.

We agree. We now consistently refer to this model as the ring-pin model throughout the text.

26. Results, paragraph entitled "ITS2 and flanking regions fold into compact structures before the formation of 5.8S": The sentence "In general, our ChemModSeq data (Fig. 6b) and follow-up primerextension data (Fig. 6c, Fig. S3a) agree best with the ring-pin structure proposed by Coleman¹⁷, which we also found to be consistent with previously published mutagenesis data³⁸⁻⁴⁰ (Fig. S3b)" refers to figures S3a and S3b, I guess that should be corrected.

This has now been corrected in the revised version.

27. Results, paragraph entitled "ITS2 and flanking regions fold into compact structures before the formation of 5.8S": I guess that the authors mean in the sentence "... and nucleotides that overlap L7 binding sites in region VI (Fig.S3), were more reactive" Rlp7 instead of L7.

Correct. Note that we have removed the section describing differences between the 27SB and 27SA₂ particle in the main text (also see below). We revisited all of the primer extension analyses that we had done on the 3' end of ITS2 and found that this region showed a lot of variability due to high background levels in the control sample. We therefore decided to remove these data (and the old Fig. S4) from the manuscript.

28. Results, paragraph entitled "ITS2 and flanking regions fold into compact structures before the formation of 5.8S": The authors say that "It was previously proposed that ITS2 undergoes a conformational change from an open to a more helical structure prior to cleavage of the 27SB pre-rRNA at site C2" and cite references 36-38. I am not sure that in references 36 and 37 such a transition was proposed.

Correct. Only the paper from Brenda Peculis's lab (Cote et al) discusses this transition.

We have made the necessary correction.

29. Results, paragraph entitled "ITS2 and flanking regions fold into compact structures before the formation of 5.8S": The authors state that "Dissociation of these factors is required for subsequent removal of ITS2 after C2 cleavage" and cite reference 13 to substantiate this statement. I guess that relates to rather extended binding interfaces between some factors and the 5.8S and 25S proximal ITS2 stems. I am not sure that there is experimental evidence clarifying the relationship between factor dissociation and initial removal of other ITS2 parts.

We agree that, there is no experimental evidence demonstrating the relationship between factor dissociation and removal of ITS2 parts. We have removed the reference in the revised version.

30. Results, paragraph entitled "ITS2 and flanking regions fold into compact structures before the formation of 5.8S": The authors conclude here that "ITS2 is generally highly structured and does not appear to undergo any major changes in secondary structure during posttranscriptional conversion of 35S to 27SB". When looking at Figure 6b and when comparing the patterns of ITS2 SHAPE reactivities in the first two lanes (35S and 27SA₂) with the 3rd lane (27SB) there are significant differences apparent. If I am not getting it wrong, in the 27SB sample there are often (especially in the region spanning nucleotide 100 to 200) regions of strongly increased nucleotide flexibility which are partially proteinase k sensitive. Were these changes re-analyzed by primer extension analyses? I think the authors point to this observation in the sentence "This again suggests that ITS2 may become slightly more flexible prior to formation of 27SB". Can we exclude that this reflects significant changes in ITS2 secondary structure which are induced by proximal or distal binding of proteins?

To validate our ChemModSeq data we performed detailed analysis of ITS2 region in 27SA₂ using primer extension. These results show that both ChemModSeq and primer extension data agree very well with each other (see new Fig. S5). Therefore, we are confident that the ChemModSeq data is accurate. We have taken a closer look at the reactivities in ITS2 of the 27SB pre-rRNA and it appears that the reactivities were generally higher in most nucleotides. Those nucleotides located in region 100-200 are almost all found in the loops or bulges of the proposed ITS2 structure. Therefore, it appears that the overall structure does not change. We now state on page 12 of the revised manuscript: *“The 27SB particle reproducibly showed higher reactivities in ITS2 in the ChemModSeq data, particularly the 100-200 region. Although this may indicate that ITS2 undergoes structural rearrangements in the Nsa2 particle, these sites were all located in bulge or loop regions (Fig. 6a), suggesting that these single-stranded regions are more effectively probed in the Nsa2 particle. It is possible that this may be related to the differences in protein composition between the Nsa2 and Rrp5 particles.”*

As mentioned in point 27, the only region where we found a high degree of variability in the primer extension data was the 3' end of ITS2. Therefore, these results made it difficult to draw conclusions about the 3' end of the ITS2 and we have therefore removed any discussion of this region from the main text and removed the primer extension gels from Figure S4.

31. Discussion, paragraph entitled "Folding of 5.8S largely takes place during the conversion of 27SA₂ to 27SB": At the end of this paragraph and in figure 8 the authors develop a model which can be understood in the sense that stable assembly of rpL17, rpL35 and rpL37 paves the way for the assembly of late B-factors and subsequently leads to processing of the 3' ITS1 and formation of 27SB (last part of this section of the discussion). When looking at the primary literature I had the impression that ITS1 processing can go on rather well in yeast expression mutants of rpL17, rpL35, rpL37 or "late B-factors". In some of them even substantial cleavage at C₂ was observed. How does that relate to the model?

We see where the confusion is coming from. Essentially what we wanted to show in Figure 8 is that these r-proteins assemble or become more stably bound during the conversion of 27SA₂ to 27SB. This is required for the assembly of the late B factors and cleavage at site C₂. The reference to A₃ cleavage and ITS1 processing should not be in this sentence. We thank the reviewer for pointing this out. In the revised manuscript we have now removed the reference to ITS1 processing.

32. Discussion, paragraph entitled "The temporal formation of (long-range) r-protein-rRNA interactions " : That paragraph recapitulates some of the statements made in the respective two parts of the results section. Specific comments on the relevant topics were given further above.

We have made substantial changes to the discussion to answers this reviewer's concerns.

Reviewer #2 (Remarks to the Author):

Eukaryotic ribosome biogenesis is an extremely complicated process. Most trans-acting factors involved have been identified. The first high-resolution structures of pre-ribosomes have been determined, providing initial insights into the function of several assembly factors.

Here the authors have addressed the dynamics of some of the restructuring events that occur during subunit assembly. They have applied chemical probing (SHAPE) on three isolated heterogeneous populations of pre-ribosomes, each amino-purified from a specific bait (Mrd1, Rrp5, and Nsa2).

There is a lot of work involved, and it is generally of good quality.

1. My main problem is that the authors are making numerous (too many in my opinion) assumptions to interpret their data (see pages 8 and 9). The description of the different categories of hits (classification) is also exceedingly complicated. The authors are making a few predictions based on their data but, regretfully, they do not validate any of them.

To simplify the interpretation of the data (and to address the concerns of Reviewer 1) we have substantially condensed the section describing the classification of hits to make it easier to follow. We also have added a number of paragraphs to the Discussion where we discuss alternative interpretations of what each category could represent. We present primer extension analyses of 5.8S and ITS2 in the particles analyzed. These analyses are in good agreement with the ChemModSeq data and therefore we feel it is reasonable to assume that most of the other changes also have been correctly described. Unless we generate high-resolution cry-EM structural analyses on the individual it will be very difficult to validate each predicted interaction that we observed in our data. This would go well beyond the scope of the current manuscript and even with the latest advances in the field we may still not get the resolution required to observe direct contacts between proteins and RNA for most regions.

2. In a subsection of their work, the authors are challenging a published model for ITS2 processing (so-called ring to hairpin model). Challenging a published model is of course completely fine, although in this case it is quite well-supported, and it has been used extensively by the authors themselves in the past. The authors should discuss whether their revised ITS2 folding model impact (?or not) their former conclusions (see Granneman et al EMBO J. 2011).

This is an excellent point. We have now added a paragraph to the discussion

where we compare our current work to the results described in the Granneman et al paper in 2011 as well as papers from the Woolford lab. This is now discussed in detail in the Discussion section of the revised manuscript.

3. In conclusion, the manuscript suffers from the highly speculative nature of the conclusions, it is very difficult to read, and altogether, the biological significance of the work is limited.

To address the reviewer's concerns, we have made significant revisions to the manuscript to improve the flow of the story. Much of the speculative nature of the paper has been reduced or moved to the Discussion section. We have also significantly condensed the results section to make the manuscript easier to read. Regarding a lack of biological significance, we respectfully disagree with this reviewer's opinion. We believe that our work presents significantly new insights into pre-60S assembly. For example, we are the first to demonstrate that 5.8S undergoes major restructuring events during a very specific stage of 60S maturation and we provide the first *in vivo* structure of ITS2. Our ChemModSeq data analyses also provide insights into how r-protein-rRNA interactions are formed during the assembly of 60S. Such details have even escaped detection in recent high-resolution cryo-EM microscopy analyses. We therefore believe our data nicely complement previous (and future) cryo-EM studies and will serve as a very useful resource for the community.

4. Comments on the methodology
-There is an important inherent limitation to the method: purifying pre-ribosomal particles from one assembly factor, and showing the RNA to be intact (by mapping its 5'-end by primer extension, and showing its size on a bioanalyzer chip) does not mean that a single type of pre-ribosome was purified. The authors are applying a single nucleotide resolution chemical mapping technique to heterogeneous populations of particles (they are doing population analysis not single particle analysis) that have, by definition, undergone various levels of restructuring events. This should be discussed.

This reviewer is correct, we cannot exclude the possibility that the population of particles is naturally heterogeneous, despite the fact that we gel purify specific pre-rRNA intermediates. We now mention this in a new paragraph in the Discussion section (Purification of ribosome assembly intermediates). We would like to stress that even though the particles might have been heterogeneous, we were still able to detect highly structured features that were present in all particles and very obvious structural rearrangements. It is very likely that what we detected were rate-limiting rRNA restructuring steps that involves the release of assembly factors or the activity of an energy-dependent enzyme. This is now addressed in the discussion in the new paragraph "Purification of ribosome assembly intermediates".

5. -Is it a problem that the Mrd1 bait is in fact a mutated construct (Mrd1

RBD5delete)?

Mrd1 is associated with precursors containing the 18S rRNA but is not required for synthesis of 25S rRNA. The Mrd1 mutant used here (Mrd1 Δ 5) accumulates modest levels of 35S but does not significantly affect the levels of 27S and 7S pre-rRNAs as well as the 25S rRNAs (Lundkvist et al 2009). It is therefore reasonable to assume that the rRNA folding steps in the 27S pre-rRNA are (largely) unaffected. FL managed to purify a small quantity of 35S from Rrp5 immunoprecipitations, which was sufficient for performing a primer extension reaction. FL compared the 35S isolated from the Rrp5-TAP IP with 35S from the Mrd1 Δ 5 strain and the results show that in the 5.8S region we did not see any noticeable differences between the samples. We therefore conclude that it is unlikely that the Mrd1 Δ 5 mutation has a significant effect on the folding of 35S and 27S pre-rRNA species. These new findings have been included in the discussion of the manuscript and the 5.8S primer extension results are presented in the new Figure S7.

6. -A new nomenclature for ribosomal proteins has been proposed based on extensive structural work and phylogeny, using it means making our field less obscure.

Although the new nomenclature is a great initiative, not all labs adhere to it and (unfortunately) the community appears to be quite divided on this matter. To make it easier for everybody to follow, we now use both nomenclatures in the text, but we maintain the old nomenclature in the figures.

Reviewer #3 (Remarks to the Author):

Here Burlacu et al have used RNA structure probing to study early RNA folding in nucleolar pre-60S particles. In particular, the authors have focused on the structural changes around ITS2 and the 5.8S rRNA. The data presented in this manuscript suggest that the 5.8S rRNA adopts its structure predominantly during the 27SA2 to 27SB transition while ITS2 adopts a compact structure early on. Overall this manuscript contains important new insights that will be very important for the field and it is suitable for publication in Nature Communications provided that the following points are addressed by the authors in a revised version.

We thank Reviewer 3 for the constructive criticism!

In Figures 1 and 2 the authors argue that the selection of Mrd1, Rrp5 and Nsa2 will result in the isolation of 35S, 27SA2 and 27SB particles only but insufficient evidence is presented to show this. Primer extensions are used only for particles isolated via tagged Rrp5 and Nsa2 to show that the associated pre-rRNAs terminate at sites A2 and BL/BS respectively. Comparative mass spectrometry is used to compare A3 and B factors only between Rrp5 and Nsa2.

1. First, the authors should provide evidence (northern blotting etc.) to show that the isolated Mrd1-associated particle indeed contains a 35S pre-rRNA. In addition, the authors should show the Mrd1-associated treated pre-rRNA in Figure 2a next to the Nsa2 and Rrp5 associated pre-rRNA species.

To address this reviewer's request, we have performed a Northern blot analysis on affinity-purified RNAs isolated with the various baits (see new Fig. S1b). These data clearly demonstrate that the Mrd1 mutant predominantly co-precipitates the 35S pre-rRNA (Figure S1b, lane 5) and we also show that the gel purified 35S is of exceptional quality (Figure S1b, lane 7). Regarding the question about Fig. 2a. The longest RNA that can be resolved on a Bioanalyzer chip is ~4kb (See the ladder in Fig. 2a). The 35S pre-rRNA is ~7kb and therefore we were not able to assess the quality of this pre-rRNA using the Bioanalyzer.

2. Second, the authors should provide SDS-PAGE analysis of all three isolated particles to address the homogeneity of each isolated particle. This will be important to ensure that a homogenous population of particles has been isolated for subsequent RNA analysis.

In the new Supplementary Figure S1 we have also included a gel pictures and a Western blot of the Rrp5, Nsa2 and Fun12 IPs that we performed (Fig. S1). We did not perform the Mrd1 MS analysis because previously we had done mass-spectrometry on Mrd1 particles (Segerstolpe et al NAR 2013) and found that pre-60S assembly factors were highly underrepresented, making a direct comparison between this particle and the other pre-60S particles very challenging.

3. This analysis will further address the following issue:\= Rrp5 is associated with early ribosome assembly intermediates including both early pre-40S particles (SSU processomes) as well as early pre-60S particles. By purifying Rrp5-associated particles, the authors clearly co-purify SSU processome components (as shown in the supplement), which presumably contain a pre-rRNA containing the 5'ETS and 18S rRNA. Alternatively, this sample may contain a 35S pre-rRNA species. The authors should provide further evidence of which other pre-rRNA is present besides 27SA2.

We completely agree with Reviewer 3 that Rrp5 will co-purify several intermediates, including small amounts of 35S (see new Fig. S1). However, to be sure that we were only analyzing a single pre-rRNA species, after chemical modification we resolved the affinity purified RNAs on an agarose gel and subsequently we gel-purified the pre-rRNAs of interest (in the case of Rrp5 we gel purified the 27SA₂ pre-rRNA). Subsequently, we performed primer extension on these purified pre-rRNAs to assess their homogeneity. The primer extension data in Fig. 4c show that the gel purified 27SA₂ has negligible amounts of 27S and 27SA₃. The gel purified 27SB pre-rRNA did not contain detectable amounts of 27SA₂. These results clearly show that we are looking mostly at a single pre-rRNA species. Of course, the purified particles (i.e. the protein composition and parts of the RNA structure) could still be heterogeneous. We have added a new paragraph to the Discussion section (Purification of ribosome assembly intermediates) where we address the issue of heterogeneity of our particles.

4. In panel 1c and 1d it is unclear which data point refers to which ribosome assembly factor and labels should be added.

We have now put the name of the assembly factors next to the data points in Fig. 1c and 1d. We hope this aids in the interpretation of the data.

5. The positions of bases with changed flexibility are very hard to follow in the three panels of Figure 3c.

This figure could be improved by using more traditional views (such as crown view etc.). Mapping the indicated bases on an associated secondary structure diagram, such as the one shown in Figure 3a, would be very helpful to correlate the position of the bases from 2D to 3D.

We agree with this reviewer that it is not easy from the images to determine where exactly the sites concentrate. We have tried various views and angles but have not been able to generate images that look very satisfying. As a result, we have now included the pymol session used to generate Figure 3c as Supplementary data. Also, in Supplementary Table S3 we now provide a new table showing the total number of PKSRP sites in each domain.

6. In the current manuscript the authors have not adequately credited a very recent cryo-EM structure of a late nuclear pre-60S particle (briefly mentioned as reference 13). This relates in particular to the (secondary) structure of ITS2. Importantly, the cryo-EM structure is in very nice agreement with the RNA structure probing data presented in the current manuscript and only agrees with the ring-pin model shown in Figure 6a. Here it would be appropriate to show the 3D structural model of ITS2 and the associated ribosome assembly factors (pdb code: 3JCT) with the RNA structure probing data.

We agree that we did not give this paper sufficient attention in our manuscript. We have now added a new panel to Figure 6 (Fig. 6d) in which the results of the chemical probing data are highlighted in the secondary structure. We tried to put these data into a 3D structure model but these results were difficult to interpret. As an alternative solution, we extracted those nucleotides in the 3JCT structure of ITS2 that are predicted to be single-stranded (i.e. not involved in Watson-Crick and Hoogsteen base-pairing interactions and not bound by proteins). These are more likely to be modified by 1M7 (although base-stacking interactions are also known to interfere with 1M7 modifications. These we did not consider). We note that in the 3JCT structure only a small part of ITS2 was resolved that includes fragments of regions I, II and VI (see new Fig. 6d). Our chemical probing results are in good agreement with this structure. Most of the highly reactive nucleotides were located in the loops where we identified single-stranded nucleotides. We have now also used this model to compare the structure of 5.8S 3JCT and the mature ribosome. These results (see new Fig. S4) show that the 3D structure of 5.8S region in the Nog2 particle is remarkably similar to the structure shown in the mature ribosome (Ben-Shem 3Å structure).

7. Lastly, there are some minor corrections that need to be addressed, such as the following typos.
 - a. On page 11: "the the observed increase"
 - b. Corrected
 - c. On page 13: "(Fig. 5d, purple nucleotides)" - the nucleotides are green
 - d. Corrected
 - e. On page 16: "our data suggests that L26 contact"
 - f. Corrected

Reviewer #4 (Remarks to the Author):

Review of Burlacu et al.

This is a highly interesting paper from the Granneman Lab presenting an analysis of 60S assembly based on affinity purification of the ribosomal complexes and RNA structure probing.

The method employed is similar to the method that Granneman and coworkers previously used to analyse the 40S subunit (Hector et al., 2014). The major finding of the study is that folding of the 5.8S rRNA occurs at a specific stage of processing, suggesting that subsequent nuclear processing events depend on the 5.8S folding to be completed. Moreover, the study provides new information about the timing of the formation of protein-RNA interactions and long range RNA-RNA interactions. These findings are clearly novel and will be of high interest in the specific field. However, in this reviewer's opinion there are some methodological issues that will have to be addressed before this paper can be published.

1. The purification of the ribosomal complexes is quite convincing and form a strong basis for the following probing experiments. While the methods used for the proteomics experiment are described in details, the description of the methods used for analysis of the SHAPE data is very short and makes it very difficult for the reader to evaluate the potential for bias in the data and indeed to reproduce the work presented in the paper. For the calculation of SHAPE reactivities the authors cite Deigan et al. 2009 and Tang et al. 2015, however, although these papers are conceptually quite similar, the described strategies also contain differences and user specified thresholds that could potentially influence the reactivity data. A detailed description of the data analysis that is performed in this paper is therefore absolutely essential before this paper is published.

We now provide more details about our data analysis pipeline in the Methods section. Furthermore, we have setup a public repository (bitbucket.org/sgrann/chemmodseqpipeline) that has a script that runs our entire pipeline over multiple processors and provides a list of all the dependencies. Moreover, we provide a PDF file in which the library construction steps as well as the bioinformatics pipeline is explained. This should enable researchers to replicate our data.

2. Based on the Deigan and Tang references, the analysis used in the manuscript must be based on a log ratio between normalized counts in the probed sample and the control. This is in contrast to the strategy that were used in the previous paper (Hector et al). The log ratio approach is more susceptible to biases introduced by differences in the coverage between the control and probed sample. Such coverage differences could arise from RNA degradation or from the probing itself. The latter is a real concern as the 1M7 reagent is used in a high concentration in this study (30mM final). If coverage differences are present between control and sample and vary between the ribosomal complexes, this

could lead to specific regions having lower or higher reactivities. This is a concern with respect to the 5.8S, where this kind of general trend is observed by the authors. As the change in 5.8S flexibility during assembly is a major claim of the paper, the authors need to rule out that differences in coverage of the probing data affects their data.

Reviewer 4 spotted an error in the Methods section. The 30 mM concentration mentioned in the Methods is incorrect. It should be 12.5 mM. This has now been corrected in the text. We carefully optimized the probing conditions and are using the lowest concentration that gave good probing results, so we are confident that our concentrations are not too high. We also did not observe a drastic signal decay in primer extension reactions, indicating that we are not over-modifying our samples. We agree that major differences in coverage profile between the control and modified samples could cause problems when using the Ding/Tang method for calculating SHAPE reactivities. However, in our data the coverage profile of the control and modified experiments looks very similar, even between biological replicates (see Figure R4.1 below). There are two pockets where we see very low coverage compared to the rest of the transcript, which was the result of natural modifications that very effectively block the reverse transcriptase. Note that for each experiment we also generate separate controls and these also show very similar coverage profiles as the modified samples. The coverage is also well over 200,000 reads per nucleotide in most cases, which is well above the recommended 5000 reads. Regarding the 5.8S data, we also performed the conventional gel-based primer extension analysis on 5.8S that confirm our ChemModSeq data. We clearly see in these primer extension data that the 5.8S region becomes significantly more compact in the Nsa2 particles, confirming our ChemModSeq results (see Fig. S3b and S3c). We also show a biological replicate in Fig. 6d. Thus, we are confident that our ChemModSeq analysis are accurate and that coverage is not an issue.

3. The obtained sequencing data is paired-end and it is therefore surprising that the authors are not using the normalization methodology that were developed in the Hector et al study, which should be more robust with respect to coverage and in addition uses a nice Poisson expectation maximization algorithm for scoring nucleotides modified or non-modified. I suggest that this method is applied for the data analysis.

That is correct. The reason why we performed paired-end sequencing was to be able to normalize the data by the coverage. We initially applied our Poisson expectation maximization (TCP_EM) algorithm to our data, but to our surprise it performed very poorly on the pre-60S data. The TCP_EM algorithm called a significantly lower number of modified nucleotides compared to the method used by Ding/Tang for calculating SHAPE reactivities (see Figure R4.2; ~5%) about 5-fold lower than the SHAPE threshold we set and about 4-5 fold less when using the Weeks threshold for intermediate (0.4) and about the same number when using the Weeks' high reactivity (0.8). We changed many different parameters of

the algorithm to see if we could improve the total number of positive calls, however, this did not yield better results. We think this may have to do with the fact that the SHAPE reactivities in the pre-60S particles were generally lower than what we observed in the 18S data, possibly because the pre-60S complexes are more dynamic in structure. Unfortunately, the person who developed the algorithm has moved on a few years ago. This is the reason why we decided to (temporarily) revert to the Ding/Tang et al SHAPE algorithm. While this work was in progress, in collaboration with Guido Sanguinetti's group we also started working on a new (and non-parametric) algorithm, that takes into consideration biological replicates and also sequence representation biases (see Selega et al *Nature Methods*, November 7th). The algorithm (BUM_HMM) performs better on the Hector et al 18S DMS data compared to the TCP_EM, SHAPE algorithms, particularly in regions that have low coverage. The algorithm does not produce a reactivity value but reports posterior probabilities (likelihood of the nucleotide being modified) and does not provide an indication of how strong the modification is. At the moment it only compares control samples to modified samples, but what we need is a way to compare modified samples from different particles so that it can produce a posterior probability of a change in flexibility. This is work in progress but we will need some time to finish the method and properly test it. Therefore, we are not yet able to apply the BUM_HMM method to the current dataset but will certainly do so in the near future. Although we agree that the Ding/Tang algorithm is not the best performing approach, from our comparative analysis we have learned that it works well on very high coverage data (which our data is) and therefore we are confident that our analysis is of high quality. We hope that this reviewer will agree that the current analysis is sufficient as we need more time to adapt the BUM_HMM model.

4. In this study, the SHAPE reactivities of the different samples are used to classify positions into modified vs non-modified based on an arbitrary cut-off. How the cut-off is chosen is not discussed, but should be clarified as it could influence the resulting data a lot. The actual cut-off used is a bit unclear from the paper, which states: "The nucleotides with 1M7 reactivities below the first quartile were considered unmodified. Nucleotides with 1M7 reactivities above the first quartile were considered modified, while the nucleotides with 1M7 reactivities above the third quartile were considered highly modified."

Clearly, the problem with these kind of analyses (a general problem in the field) is that we have to set arbitrary thresholds for calling modified nucleotides. The reason why we considered reactivities above the first quartile as "modified" is because at this threshold most of the differences in SHAPE reactivities were higher or equal to 0.3, which according to a paper from the Weeks lab (Rice et al RNA 2014) can be considered as strong changes in SHAPE reactivity. This has now been written more clearly in the results section in the main text on page 7.

5. In the main text it is stated that 1940 nucleotides were found to react with 1M7,

but in the Table S1, samples have 1294 to 1405 out of the 3786 nucleotides that are called modified. Neither of these numbers seem to be close to either $\frac{1}{4}$ of the data points or $\frac{3}{4}$ of the data points, which should be explained. Again the previously developed method using Poisson expectation maximization should be superior to the one use here.

We thank Reviewer 4 for pointing out this mistake. We now included the correct number on page 8 of the revised manuscript. An average of 1284 nucleotides reacted with 1M7 over the 5 types of pre-rRNA samples analyzed.

- The authors do not attempt to validate the quality of their structure probing data. As a minimum, it would be relevant to test if the authors reactivity data significantly correlate with the crystal structure basepair annotation and when the authors identify sets of nucleotides that seem to change their as a result of protein binding, it would be relevant to demonstrate that this set is significantly closer in space to protein in the relevant ribosomal complexes than other positions.

We had done a number of analyses to determine how well our SHAPE data correlates with the crystal structure data (Ben-Shem et al 80S structure data). We extracted nucleotides in the crystal structure that were single stranded (i.e. were not involved in Watson-Crick base-pairing interactions) and we also selected nucleotides that were single-stranded and also not contacted by proteins. For each domain in the 25S rRNA regions, we then generated receiver operator characteristic curves (ROC) where we compared the SHAPE reactivity values with the crystals structure data as binary classifier. We then calculated the area under the ROC curves (AUCs) for each domain in the 25S sequence. Values higher than 0.5 suggest that our data agrees well with the crystal structure data.

		Single-stranded regions only				
domain	25S region	35S	27SA2	27SB	27SA2_PK	27SB_PK
1	1 to 635	0.633	0.6	0.567	0.54	0.53
2	661 to 1431	0.712	0.65	0.63	0.64	0.59
3	1452 to 1879	0.6	0.58	0.63	0.57	0.54
4	1911 to 2332	0.63	0.54	0.52	0.53	0.55
5	2400 to 2979	0.6	0.59	0.62	0.63	0.63
6	2994 to 3396	0.654	0.59	0.52	0.56	0.54

domain	25S region	Also considering protein-binding sites				
		35S	27SA2	27SB	27SA2_PK	27SB_PK
1	1 to 635	0.653	0.589	0.599	0.527	0.522
2	661 to 1431	0.647	0.637	0.667	0.644	0.622
3	1452 to 1879	0.585	0.525	0.635	0.483	0.4722
4	1911 to 2332	0.613	0.518	0.475	0.496	0.551
5	2400 to 2979	0.516	0.535	0.6	0.567	0.578
6	2994 to 3396	0.558	0.491	0.468	0.462	0.447

As is evident from the tables, for several domains we find AUCs lower than 0.5 in the particles, suggesting that the SHAPE data profile does not reveal a structure that is very similar to the crystal structure (or performs poorly compared to the crystal structure). However, this is really not surprising, since we are looking at one of the earliest intermediates that not only have a different r-protein composition, but are also associated with dozens of assembly factors that do not interact with the mature ribosome. This will greatly influence the reactivity pattern. Therefore, we do not believe these results are really informative of the quality of our SHAPE data or how similar the structure in our particles is compared to the crystal structure as there are too many differences in protein composition. What is interesting is that some domains appear more similar to the crystal structure than others (such as domains I and II), which may indicate an order in which certain domains are folded. However, this conclusion is highly speculative. Therefore, we would prefer not to include these data in the manuscript.

7. In Hector et al. *in vivo* probing with NAI is performed and inclusion of this kind of data for the non-proteinase K samples would certainly improve the paper, as it is unclear how purification affects the RNA structure of the different ribosomal complexes.

We initially used NAI to perform *in vivo* probing, however, to our surprise, the incubation with NAI appeared to result in dissociation of the pre-ribosomal complexes that we were studying. We spent quite some time optimizing the NAI

probing protocol to make sure we were not using too much reagent. However, we could not affinity purify sufficient material for structure probing analyses. For some reason this was not a big issue for the more stable pre-40S particles analyzed in our NAR paper. We now discuss these findings in a new paragraph in the discussion (Purification of ribosome assembly intermediates). For the NAI probing the cells need to be centrifuged and resuspended in a much smaller volume. It is possible that this stresses the cells and results in turnover of early intermediates.

8. Minor points.

In figure 3b, low coverage should be removed, because otherwise the figure is very difficult to interpret.

We agree with Reviewer 4 to a point, but if we were to remove the low coverage data from this figure, it might give the reader the impression that a lot more structural rearrangements were detected in some of the domains. For example, in domain VI, which is located at the 3' end of the 25S rRNA sequence, 40% of the nucleotides had less than 5 drop-off counts. If we would remove these data, then the reader might get the impression that a lot more nucleotides showed changes in flexibility. We feel that it is important to present all the data, including the negative data.

9. The yeast rRNA crystal structure paper should be cited in figure texts.

We now cite the 2011 Science paper from the Yusupov lab in all figure legends as well as the main text.

10. The method used here so different from Hector et al. that it does not make sense to use the ChemModSeq name about the experiments.

The protocol we used for making the libraries is identical to the ChemModSeq protocol. The only difference between this work and our previously published work is that we did not use the TCP_EM algorithm for our data analyses. The ChemModSeq library preparation method is in several aspects different from what other groups use and therefore we would prefer not to use a different name at this point as this might cause confusion. For example, we were the first to do paired-end sequencing for quantifying primer extension stops.

11. Figure 2b is not especially intuitive and should be modified.

We agree. We have made extensive changes to Fig. 2b as well as the main text to simplify the interpretation of the various chemical modification states. We hope this makes it easier for the reader to follow the interpretation of our results.

12. Figure 1c: I am not completely convinced that it is a good idea to compare spectrum counts for different proteins. Would it make sense to compare the

spectrum count for the same protein in the two pull downs and then subsequently evaluate if they are A3 or B?

Correct, the Reviewer's suggestion would be the best approach if the two particles had a very similar composition and if only a handful of proteins would differ in abundance. However, there is quite a big difference in the protein composition between the Rrp5 and Nsa2 particles. The Rrp5 particle also contains many proteins involved in 40S biogenesis. We reasoned that this difference in complexity could influence the spectra count for individual proteins found in each particle. Therefore, we are not comfortable comparing the levels of the same proteins between two different experiments.

13. The title is "High-throughput RNA structure probing reveals critical folding events during early stages of ribosome biogenesis in yeast". Here, I feel that critical should be removed as the authors not really demonstrate the folding is a critical event.

We have identified a folding event in the 5.8S region of the pre-rRNA during nucleolar ribosome assembly. Furthermore, proteins that bind to 5.8S region and are responsible for its folding (such as L35 and L37) are essential for ribosome function. This motivates the use of "critical" in our title.

Fig. R4.1. Coverage profiles of 27SA₂ and 5.8S ChemModSeq data. DMSO_1 and DMSO_2 are biological replicates of two negative controls. 1M1_1 and 1M7_2 are biological replicates of treated samples. Shown are the results for the Rrp5 particle data (a-c) and the proteinase K treated Rrp5 particles (b-d). The results demonstrate that the coverage profile between the two samples (PK and particle) is very similar and the vast majority of nucleotides are covered by at least 200.000 reads.

Fig. R4.2. SHAPE reactivity algorithm calls a significantly higher number of modified nucleotides. The boxplot displays the distributions the fraction of nucleotides called modified (y-axis) in all the samples analyzed (35S, 27SA₂, 27SB and all deproteinized samples) at various SHAPE reactivity thresholds and in the TCP_EM data (x-axis). The results show that at our threshold used for calling modified nucleotides we call more than five times the amount of modified nucleotides compared to the TCP_EM algorithm, whereas at the Weeks threshold of 0.4, this number is around 4-fold. Clearly, the TCP_EM algorithm is too conservative.

Reviewers' comments:

Reviewer #1 (Remarks to the Author):

In my opinion, the revised version of the manuscript benefitted from several improvements and the major points of criticism raised by the reviewers could be addressed in a satisfactory way.

Reviewer #2 (Remarks to the Author):

Some of the original assumptions of this work are still there, but the authors have improved the readability of their manuscript. They are also more cautious in their description and interpretation. The description of the different classes of hits (which was overly complicated) has been greatly simplified – for the better. Much appreciated is that sufficient alternative interpretations of their chemical probing dataset is now offered in the discussion.

A major assumption in this work (e.g. Fig 4, Fig 7) is that 'altered 1M7 sensitivity of specific nucleotides in proteinase K-treated samples reflect rRNA restructuring events which are directly dependent upon ribosomal protein binding at these sites'. This, of course, is plausible. Just as likely a possibility is that such differences in 1M7 sensitivity reflect indirect effects, i.e. mid- to long-range effects with allosteric transmission caused by loss of binding of an unrelated factor/ribosomal protein in the neighbourhood of the modified nucleotides. This latter 'very real' alternative explanation is now better acknowledged by the authors (discussion section).

If comfortable with this assumption (important has it has major consequences for the interpretation of half of the manuscript, see for ex. Fig 4, Fig 7 etc) then the paper is sound, and it provides sufficient useful information (e.g. enrichment of PK-sensitive sites at the subunit solvent side of the subunit in Fig 3c; timing of ITS2 proximal stem formation in Fig 5; revised ITS2 structure in Fig 6, etc).

!!! Please use the new nomenclature for r-proteins in your figures (that's what people are looking at, and will remember), not just in your text (Figures 4 and 7 for example). Thank you. Most Labs in the field have started to use this new nomenclature whose rationale integrates both structural and evolutionary considerations (unlike previous ones which were purely historical).

Reviewer #3 (Remarks to the Author):

Burlacu et al. have provided a very detailed response to all reviewers and in my opinion, their comments address all previous concerns. This manuscript can be published in Nature Communications.

Reviewer #4 (Remarks to the Author):

Review of Burlacu et al revised version.

Granneman and coworkers have made some changes to the manuscript that have improved the manuscript substantially, but there are also some of my concerns, which has not been addressed appropriately.

The SHAPE data analysis is now very well described and it is great that the authors has made their scripts available, thereby making their analysis reproducible and transparent.

The authors still arbitrarily set the cut-off between modified/flexible and not modified/constrained to the first quantile for each dataset. I looked at the data in supplementary table 2 and I still cannot get the numbers to add up. For instance, for the X27SA2 data, 1294 positions are called flexible, 2223 are called constrained and 269 are not analysed. I may be missing something, but it remains unclear to me how this can be the result of the described quantile cut-off procedure. The authors still do not explain in the paper why the first quantile is a relevant cut-off. In the rebuttal letter, they state: "The reason why we considered reactivities above the first quartile as "modified" is because at this threshold most of the differences in SHAPE reactivities were higher or equal to 0.3, which according to a paper from the Weeks lab (Rice et al RNA 2014) can be considered as strong changes in SHAPE reactivity."

I cannot see how a high level of differential SHAPE values can be used as an argument for choosing the cut-off. This is also strange because the authors do not use the differential SHAPE values in their analysis. It may be that using the differential SHAPE values for assigning whether there is a change from one state to another would be a more robust strategy than the strategy used in the manuscript (which is sensitive to the cut-off chosen). Moreover, that most positions have high differential SHAPE values (fig S2) is to me a concern and indicates that there may be some issues, which makes it difficult to compare the different datasets.

One such issue could be differences in the coverage among samples. In figure R4.1, the authors show the coverage for some of their samples. In the rebuttal letter, they state:

"We agree that major differences in coverage profile between the control and modified samples could cause problems when using the Ding/Tang method for calculating SHAPE reactivities. However, in our data the coverage profile of the control and modified experiments looks very similar, even between biological replicates (see Figure R4.1 below)."

It is true that the overall shape of the obtained coverage is somewhat similar, but the plots also show that there are regions that have quite large differences in coverage between samples across the rRNA. If the coverage should have no influence on the data, the curves should be scaled versions of each other, which is far from the case. Likewise, differences in coverage between the experiments for the different pre-60S particles could also affect the result of this study. I therefore respectfully disagree with the authors when they in the rebuttal letter state that they are confident that coverage is not an issue for their results.

Minor points:

1. On page 22 in the rebuttal letter the authors discuss the TCP_EM, Ding/Tang and their recently published BUM_HMM. I agree that it seems that the TCP_EM algorithm scores too few modified nucleotides and it is possible that implementation of the new BUM_HMM algorithm could improve this manuscript. The authors speculate about why TCP_EM performs poorly:

"We think this may have to do with the fact that the SHAPE reactivities in the pre-60S particles were generally lower than what we observed in the 18S data, possibly because the pre-60S complexes are more dynamic in structure."

If pre-60S was more dynamic, one would expect higher reactivities?

2. Figure 3B: In my opinion the figure in its current form gives the false impression that domain VI has much lower fraction of nucleotides that are unchanged. The reader must rescale the bars to see this is not the case.

3. In the manuscript, it would be relevant to clearly outline the differences between this study and the Hector et al paper and explain why the TCP_EM algorithm was not used. Especially, if the ChemModSeq name is reused.

4. It would be relevant to include the validation of structure signal, which is presented in the rebuttal letter, in the manuscript, perhaps just for the entire 25S region. The signal is not great, but I see the authors' point with the assembly factors affecting the signal.

Reviewers' comments:

Reviewer #1 (Remarks to the Author):

In my opinion, the revised version of the manuscript benefitted from several improvements and the major points of criticism raised by the reviewers could be addressed in a satisfactory way.

Reviewer 1's constructive criticism enabled us to significantly improve the manuscript!

Reviewer #2 (Remarks to the Author):

1. Some of the original assumptions of this work are still there, but the authors have improved the readability of their manuscript. They are also more cautious in their description and interpretation. The description of the different classes of hits (which was overly complicated) has been greatly simplified – for the better. Much appreciated is that sufficient alternative interpretations of their chemical probing dataset is now offered in the discussion.

A major assumption in this work (e.g. Fig 4, Fig 7) is that 'altered 1M7 sensitivity of specific nucleotides in proteinase K-treated samples reflect rRNA restructuring events which are directly dependent upon ribosomal protein binding at these sites'. This, of course, is plausible.

Just as likely a possibility is that such differences in 1M7 sensitivity reflect indirect effects, i.e. mid- to long-range effects with allosteric transmission caused by loss of binding of an unrelated factor/ribosomal protein in the neighbourhood of the modified nucleotides. This latter 'very real' alternative explanation is now better acknowledged by the authors (discussion section).

If comfortable with this assumption (important has it has major consequences for the interpretation of half of the manuscript, see for ex. Fig 4, Fig 7 etc) then the paper is sound, and it provides sufficient useful information (e.g. enrichment of PK-sensitive sites at the subunit solvent side of the subunit in Fig 3c; timing of ITS2 proximal stem formation in Fig 5; revised ITS2 structure in Fig 6, etc).

We thank Reviewer 2 for his/her constructive comments and are pleased that this reviewer believes that the paper has been substantially improved.

2. !!! Please use the new nomenclature for r-proteins in your figures (that's what people are looking at, and will remember), not just in your text (Figures 4 and 7 for example). Thank you. Most Labs in the field have started to use this new nomenclature whose rational integrates both structural and evolutionary considerations (unlike previous ones which were purely historical).

As requested we now use both the old and the new nomenclature in both the text and the Figures.

Reviewer #3 (Remarks to the Author):

Burlacu et al. have provided a very detailed response to all reviewers and in my opinion, their comments address all previous concerns. This manuscript can be published in Nature Communications.

Fantastic, thank you!

Reviewer #4 (Remarks to the Author):

Review of Burlacu et al revised version.

Granneman and coworkers have made some changes to the manuscript that have improved the manuscript substantially, but there are also some of my concerns, which has not been addressed appropriately.

The SHAPE data analysis is now very well described and it is great that the authors has made their scripts available, thereby making their analysis reproducible and transparent.

1. The authors still arbitrarily set the cut-off between modified/flexible and not modified/constrained to the first quantile for each dataset. I looked at the data in supplementary table 2 and I still cannot get the numbers to add up. For instance, for the X27SA2 data, 1294 position are called flexible, 2223 are called constrained and 269 are not analysed. I may be missing something, but it remains unclear to me how this can be the result of the described quantile cut-off procedure.

The reason for why we used quartiles and not an arbitrary cutoff (for example the 0.4 cutoff frequently used by the Weeks lab), is because there can sometimes be variability in the 1M7 reactivity between particles. For example, it is possible that in some particles most of the reactivity values are between 0 and 1.5 whereas in other samples it could be between 0 and 2.5. Thus, we reasoned that it would be better to use quartiles (i.e. select an arbitrary threshold for each individual sample based on the median of the range) than selecting the same threshold for each sample. So essentially what we did is to quantile normalize our data to make sure all our values have a similar distribution. This is a standard (and widely accepted) way of normalizing microarray data. We picked the first quartile of all values above zero as threshold and all nucleotides with values above this threshold were called modified. When we take all the SHAPE reactivities for the 27SA2 data that were higher than 0 and calculate the first quartile for this range, then we find that 1294 positions have SHAPE reactivities higher than the first quartile, 2223 are lower (0's and lower than the first quartile) and the rest does not have sufficient coverage. We now explain this in more detail in the Results and Methods section.

2. The authors still do not explain in the paper why the first quantile is a relevant

cut-off. In the rebuttal letter, they state: “The reason why we considered reactivities above the first quartile as “modified” is because at this threshold most of the differences in SHAPE reactivities were higher or equal to 0.3, which according to a paper from the Weeks lab (Rice et al RNA 2014) can be considered as strong changes in SHAPE reactivity.” I cannot see how a high level of differential SHAPE values can be used as an argument for choosing the cut-off. This is also strange because the authors do not use the differential SHAPE values in their analysis. It may be that using the differential SHAPE values for assigning whether there is a change from one state to another would be a more robust strategy than the strategy used in the manuscript (which is sensitive to the cut-off chosen).

We apologize for still not making this sufficiently clear in the text. We have now revised the text to make our reasoning much clearer.

This reviewer is correct; we did not use the differences in SHAPE reactivities to classify each nucleotide. The reason for this is also explained in point 1. If we would simply take the differential SHAPE reactivity and assuming there was variability in the range or distribution of SHAPE reactivity values, then we might end up with a lot of false positives or false negatives. It is important that we first do the normalization step. However, to test whether our threshold is actually appropriate for our data, we calculated the differences in SHAPE reactivity for each classified nucleotide that showed changes in flexibility in the particles (i.e. those nucleotides belonging to classes 5-12). This showed that the differences in SHAPE reactivity was generally well above the threshold of 0.3, which is considered a high differential reactivity by the Weeks lab. This gave us the confidence that with this threshold the vast majority of nucleotides that we discussed in the paper (including the PKSRP sites, etc), have a high differential SHAPE reactivity. So calculating the differential SHAPE reactivity was simply our way of testing whether our threshold was adequate. This is now discussed in more detail in the text. Thus, high level of differential SHAPE values was not used “as an argument for choosing the cut-off” The corresponding phrase was corrected.

3. Moreover, that most positions have high differential SHAPE values (fig S2) is to me a concern and indicates that there may be some issues, which makes it difficult to compare the different datasets.

We apologize for not providing all necessary information in the legend for Fig. S2. This figure does not show differential 1M7 reactivities for *all* the nucleotide positions but only those positions that showed a change in flexibility (nucleotides from categories 5-12). Thus, these nucleotides generally showed a very high differential SHAPE reactivity, which was very encouraging and gave us the confidence that our arbitrary threshold was adequate. We have now changed the text in the Figure legend to make it more clear. We agree with Reviewer 4 that it would have been a concern if most positions showed high differential 1M7 reactivities. This was not the case.

4. One such issue could be differences in the coverage among samples. In figure R4.1, the authors show the coverage for some of their samples. In the rebuttal letter, they state:

“We agree that major differences in coverage profile between the control and modified samples could cause problems when using the Ding/Tang method for calculating SHAPE reactivities. However, in our data the coverage profile of the control and modified experiments looks very similar, even between biological replicates (see Figure R4.1 below).”

It is true that the overall shape of the obtained coverage is somewhat similar, but the plots also show that there are regions that have quite large differences in coverage between samples across the rRNA. If the coverage should have no influence on the data, the curves should be scaled versions of each other, which is far from the case. Likewise, differences in coverage between the experiments for the different per-60S particles could also affect the result of this study. I therefore respectfully disagree with the authors when they in the rebuttal letter state that they are confident that coverage is not an issue for their results.

This reviewer is correct that the coverage profile between particles is slightly different, which is likely to be due to technical variability. It is technically challenging to get identical coverage profiles for each RNASeq library (which ChemModSeq libraries essentially are). To minimize noise introduced by variability in coverage between particles, we included negative controls for each pre-rRNA intermediate that were prepared and analyzed in parallel with their corresponding modified sample. Furthermore, we performed every experiment at least twice. We agree that differences in coverage between the 1M7 and DMSO samples within the same experiment could affect the outcome when determining SHAPE reactivities, but this is not true for our case. The profiles of samples of the same experiment are very similar, although not identical. In fact, we observed greater differences in coverage profiles between sample pairs (1M7_1 vs 1M7_2; DMSO-1 vs DMSO_2) in different experiments rather than between 1M7 and DMSO within the same experiment. Thus, we cannot see how this issue could significantly influence our results. Also, when calculating the SHAPE reactivities, we *only* used the negative controls that were generated in parallel with the modified samples.

Minor points:

1. On page 22 in the rebuttal letter the authors discuss the TCP_EM, Ding/Tang and their recently published BUM_HMM. I agree that it seem that the TCP_EM algorithm scores too few modified nucleotides and it is possible that implementation of the new BUM_HMM algorithm could improve this manuscript. The authors speculate about why TCP_EM performs poorly:
“We think this may have to do with the fact that the SHAPE reactivities in the pre-60S particles were generally lower than what we observed in the 18S data, possibly because the pre-60S complexes are more dynamic in structure.”
If pre-60S was more dynamic, one would expect higher reactivities?

We admit that it is difficult to predict how dynamic structures would influence reactivity. Our reasoning is as follows: If a structure is more dynamic, then the obtained reactivities reflect an average of all the possible conformations. One would expect to get higher number of reactive nucleotides, but their reactivity values would generally be lower because of the presence of different conformations. If a structure is very stable, then single stranded regions might be more effectively probed and have higher reactivity values.

2. Figure 3B: In my opinion the figure in its current form gives the false impression that domain VI has much lower fraction of nucleotides that are unchanged. The reader must rescale the bars to see this is not the case.

The problem with domain VI is that we don't know whether it has a much higher or lower number of nucleotides that are unchanged simply because about 40% of the nucleotides have low coverage and therefore were not analyzed. It is entirely possible that the vast majority of these nucleotides are constrained and therefore the ration could change significantly. If we were to remove the low-coverage counts from this figure, then we might give the reader the false impression that domain VI has a much higher fraction of flexible and PK-insensitive nucleotides compared to the rest of the domains. Hence, we would prefer to leave the figure as it is. However, we agree that it may give the reader a false impression and have therefore added a new sentence on page 9 of the main text:

“Although it seems that Domain VI, located in the 3' end of the 25S rRNA region, had a much smaller number of nucleotides classified as non-reactive, this Domain also had a higher number of nucleotides with low coverage. Therefore, it is possible that our analysis underestimates the total number of constrained nucleotides in this domain”

We hope that this is a reasonable compromise.

3. In the manuscript, it would be relevant to clearly outline the differences between this study and the Hector et al paper and explain why the TCP_EM algorithm was not used. Especially, if the ChemModSeq name is reused.

The only difference between this study and the Hector et al study is that we did not use the TCP_EM algorithm. The rest (such as modification protocol and sequencing library preparation) is identical. The ChemModSeq name refers to the technique, rather than the data analysis methods and therefore we feel that is OK to keep the name. To reiterate, there are major differences between our library preparation protocols and the ones recently published so therefore we feel it is essential that we keep the name.

We do now mention in the Methods section why we decided to use the SHAPE reactivity approach and not the TCP_EM algorithm.

4. It would be relevant to include the validation of structure signal, which is

presented in the rebuttal letter, in the manuscript, perhaps just for the entire 25S region. The signal is not great, but I see the authors' point with the assembly factors affecting the signal.

We now discuss this on page 9 of the revised manuscript. The AUC data has been included in the new Supplementary Figure 4.

Reviewers' comments:

Reviewer #4 (Remarks to the Author):

Review of Burlacu et al revised version 2.

Granneman and coworkers have made some changes to the manuscript, which makes the methods used for the SHAPE data analysis more clear. I think this has improved the manuscript and addresses some of my concerns, but in my view, the analysis of the probing data is still not optimal and I am still concerned that some of the results could potentially be influenced by differences in coverage between samples and the arbitrary quantile cut-off used to differentiate modified/flexible and not modified/constrained positions.

Comments:

1. The quantile cut-off strategy used to differentiate modified/flexible and not modified/constrained is explained more carefully in the new revised version, making it possible to follow the analysis.
2. The quantile cut-off implemented in the paper ignores the zero values. Two of the samples have around 100 fewer zero values than the others, which will affect the number of positions scored as flexible for the different samples.
3. In the rebuttal letter, the authors state: "So essentially what we did is to quantile normalize our data to make sure all our values have a similar distribution.". This is misguided, as quantile normalization has nothing whatsoever to do with the quantile cut-off used here (except for the word quantile). However, I agree that it often is important to scale this type of data. The authors in this paper use the Weeks lab's 2-8 % method for this. In this case, the scaling is not as important, because the authors use a quantile based cut-off to divide positions in modified/flexible and not modified/constrained. The 2-8 % scaling therefore only serves to remove the top 2% most reactive positions from the dataset, which should be clarified in the text.
4. The scaling would have been important, if the authors used differential SHAPE values to score the positions that changes their flexibility between two samples. Using a cut-off on the differential SHAPE values would likely be less biased than applying the cutoff to the individual samples.
5. The authors acknowledge that differences in coverage between the 1M7 and DMSO samples within the same experiment affect the calculation of SHAPE reactivities. They furthermore state that the coverage of their experiments and corresponding controls are very similar and that coverage therefore cannot affect their results. Given the coverage curves provided in the rebuttal letter 1, I am not as convinced as the authors. I therefore suggest that the authors reanalyse the data with a method taking the coverage into account. Alternatively, the authors could provide coverage plots for each pair of samples/control that were used together for calculation of SHAPE values for the different experiments in a format so that the different lines can be easily distinguished. This will make it possible to compare the relevant samples and determine if the shapes of the coverage plots are indeed very similar.

Reviewer #5 (Remarks to the Author):

In this paper the authors apply ChemModSeq to the study of pre-ribosomal RNA folding. This report deals with a specific question raised by previous reviewers concerning the data analysis method applied to compare SHAPE reactivities between rRNAs in the presence or absence of ribosomal proteins.

The experimental protocol appears to be as follows: specific rRNAs are pulled down from lysed cell cultures and then modified with either 1M7 or a DMSO control. The modification was also performed on rRNAs treated with proteinase K to probe the RNAs in a protein-absent state. For each condition (with and without proteins), the ChemModSeq experimental pipeline was used to generate sequencing libraries (specifically random priming reverse transcription followed by Illumina library preparation). After sequencing, reactivities were calculated and compared between the two conditions. Several comments from the reviewer concerned exactly how this comparison is made.

Specifically, reactivity data was calculated and normalized according the Weeks lab 2%/8% rule (throwing out the top 2% of reactivity values as outliers and using the next 8% to calculate a normalization constant to apply to the entire reactivity data set.) The authors then mention that differences in read coverage can cause reactivity distributions to be different and thus to control for this they classified their reactivity values as being low (i.e. not modified) or high (modified) using a quartile classification scheme where everything above the 1st quartile was considered modified. Using these classified reactivity values (essentially a binary representation of the reactivity profile for each condition), the authors then compare between the with and without protein cases to make conclusions about protein binding. The major motivation for this is the authors claim that differences in coverage can cause differences in reactivity values which should presumably be less than when classified this way.

The previous reviewer's concern are twofold - i) that there is a bias with first applying a cutoff to individual samples before making a comparison and ii) that differential coverage should somehow be taken account in the analysis. Overall we agree with this reviewer. After carefully reviewing the methods section of the manuscript, the previous papers (i.e. the original ChemModSeq paper), and the code used to process the data in this manuscript we have several suggestions for alternative analyses that can be used to support the conclusions of the manuscript:

1.) There are ways to account for coverage differences in calculating SHAPE reactivities. This was derived in the original SHAPE + Sequencing data analysis paper (Aviran et al. PNAS 2011 doi:10.1073/pnas.1106541108) and subsequently made more understandable in a follow up paper (Aviran et al. Allerton Conference Proceedings 2011 doi:10.1109/Allerton.2011.6120379). In these papers, reactivities are considered to be a probability distribution over the RNA molecule that a specific site would be modified (so called theta reactivities). Theta reactivities do assume a Poisson distribution of the number of modifications per molecule, but the modification rate is one of the parameters that is fit using the maximum likelihood approach to deriving the reactivities. The point here is that the theta calculation inherently normalizes for differences in coverage. However it may not be possible to easily extend this analysis to the case of random priming which is used in the ChemModSeq approach.

If that is the case, then what could be a workaround would be to normalize all reactivities to sum to 1 over the entire RNA molecule (like theta's are normalized) which should naturally remove biases associated with coverage. (Note that 2%/8% normalization does not do this.) We thus propose to perform this normalization and then repeat the analysis performed in the paper as a backup to make sure conclusions are robust. (Note that this comparison is valid for RNAs of the same length but should not be used to compare between RNAs of different lengths since length affects the normalization.)

However, in reviewing the code used to calculate SHAPE reactivities we realized that several ad-hoc data analysis steps were performed which may need to be adjusted before trying this pseudo-theta normalization approach. Specifically from analyzing the code it seems the following steps are

performed:

- a. Remove mapped data points with low coverage and low drop-off.
- b. Then to deal with replicates, sum all the reads in each channel, or average (depending on options used.) The text seems to indicate that summation was used.
- c. Then calculate reactivities according to $\text{reactivity} = \log(x+1)/\text{avg}(x) - \log(y+1)/\text{avg}(y)$ where X is what was calculated in step b. for the positive channel and Y for negative channel.
- d. Then remove anything above $1.5 \cdot \text{IQR}$ before dividing everything by the NEW 10th percentile for only positive reactivities. This ends up being a not quite straightforward mix of box-plot and 2%/8% percentile that is hard to justify.
- e. Then optionally add a 2%/8% normalization on previous step's result, but only on positive values.

It appears that step d. was not reported in the methods section of this paper and data should be re-analyzed WITHOUT steps d and a prior to doing the pseudo-theta normalization. Additionally we agree with the previous reviewer that zeros should be taken into account in the analysis.

2.) The Weeks lab has already published a rigorous method for detecting protein binding using differential SHAPE reactivities (Smola et al. *Biochemistry* 2015 doi:10.1021/acs.biochem.5b00977) - see Figure 1. This analysis takes into account variability between data sets and uses a Z-score cutoff in the comparison between conditions to call positions that have different SHAPE reactivities due to protein binding. The authors should apply this method to their datasets to verify their conclusions. Note that it may be necessary to perform the normalization from suggestion 1 before performing this analysis and this should be done on reactivities that have not been classified (i.e. continuous scale reactivities).

3.) If the authors insist on performing their classification scheme before making the comparison, then we suggest using the previously published method from SeqFold (Ouyang et al. *Genome Research* 2013 doi:10.1101/gr.138545.112) which performs a similar classification from reactivities to binary values (their 'structure preference profile') using more rigorous statistics than quartile classification. At least this analysis would confirm that the quartile classification is not confounding data interpretation.

General Comments:

In general the impression from reading the methods section was that the authors were avoiding using the richness of the continuum reactivity scale and it was not clear why. Hopefully these suggestions will put their analysis on a firmer footing.

In addition, the data analysis method section was very confusing with mentions of pipelines used in the original ChemModSeq paper AND the new Nature Methods paper but the code doesn't seem to follow the new Nature Methods paper analysis as indicated. Rather, the code appears to follow the ChemModSeq analysis despite the authors saying that they did NOT use it in this present work. This needs to be clarified.

Furthermore, the authors state in the new Nature Methods paper that:

"Our statistical pipeline addresses a number of important problems in the analysis of high-throughput RNA secondary structure probing data. First, it explicitly models the biological variability of the data, providing a statistical basis for determining the significance of the observed signal. As such, it removes the need to set arbitrary thresholds and perform extensive postprocessing of the analysis results, yielding a clean and statistically interpretable pipeline."

The last sentence begs the question as to why the authors are doing the various processing and thresholding steps reported here. Perhaps they should rely on the method described in their Nature Methods paper to also repeat these analyses to make sure their conclusions are robust.

Dear Reviewer #5.

We very much appreciate your feedback on our manuscript. Your comments were very useful and constructive. We have reanalyzed our data using the suggestions you brought forward and have made a number of changes to the manuscript to include the new analyses. These changes are discussed below. Importantly, both the old and the new data analysis methods support our general conclusions.

Reviewer #5 (Remarks to the Author):

In this paper the authors apply ChemModSeq to the study of pre-ribosomal RNA folding. This report deals with a specific question raised by previous reviewers concerning the data analysis method applied to compare SHAPE reactivities between rRNAs in the presence or absence of ribosomal proteins.

The experimental protocol appears to be as follows: specific rRNAs are pulled down from lysed cell cultures and then modified with either 1M7 or a DMSO control. The modification was also performed on rRNAs treated with proteinase K to probe the RNAs in a protein-absent state. For each condition (with and without proteins), the ChemModSeq experimental pipeline was used to generate sequencing libraries (specifically random priming reverse transcription followed by Illumina library preparation). After sequencing, reactivities were calculated and compared between the two conditions. Several comments from the reviewer concerned exactly how this comparison is made.

1. Specifically, reactivity data was calculated and normalized according the Weeks lab 2%/8% rule (throwing out the top 2% of reactivity values as outliers and using the next 8% to calculate a normalization constant to apply to the entire reactivity data set.)

The authors then mention that differences in read coverage can cause reactivity distributions to be different and thus to control for this they classified their reactivity values as being low (i.e. not modified) or high (modified) using a quartile classification scheme where everything above the 1st quartile was considered modified.

The reason we used the quartile classification to set a threshold for modification was not because we wanted to correct for differences in *read coverage* but to correct for differences in *1M7 reactivity*. We stated this on page 7 of the original manuscript (also see response to point 2). During the course of this work we used several different batches of 1M7 and it is possible that some were more active than others. Some particles may therefore have higher reactivity values than others. We reasoned that doing it this way would remove this possible difference in reactivity.

2. Using these classified reactivity values (essentially a binary representation of the reactivity profile for each condition), the authors then compare between the with

and without protein cases to make conclusions about protein binding. The major motivation for this is the authors claim that differences in coverage can cause differences in reactivity values which should presumably be less than when classified this way.

Just to clarify, on page 7 of the original manuscript we made the following statement:

“To compare modification patterns of different particles and to control for variability in the level of 1M7 modification between different particles, we defined an arbitrary reactivity threshold based on quartiles of positive SHAPE reactivity values. Nucleotides with 1M7 reactivities above the first quartile were considered modified.”

So we did not use this classification for correcting for differences in total coverage between the samples.

3. The previous reviewer's concern are twofold - i) that there is a bias with first applying a cutoff to individual samples before making a comparison and ii) that differential coverage should somehow be taken account in the analysis. Overall we agree with this reviewer. After carefully reviewing the methods section of the manuscript, the previous papers (i.e. the original ChemModSeq paper), and the code used to process the data in this manuscript we have several suggestions for alternative analyses that can be used to support the conclusions of the manuscript:

- 1.) There are ways to account for coverage differences in calculating SHAPE reactivities. This was derived in the original SHAPE + Sequencing data analysis paper (Aviran et al. PNAS 2011 doi:10.1073/pnas.1106541108) and subsequently made more understandable in a follow up paper (Aviran et al. Allerton Conference Proceedings 2011 doi:10.1109/Allerton.2011.6120379). In these papers, reactivities are considered to be a probability distribution over the RNA molecule that a specific site would be modified (so called theta reactivities). Theta reactivities do assume a Poisson distribution of the number of modifications per molecule, but the modification rate is one of the parameters that is fit using the maximum likelihood approach to deriving the reactivities. The point here is that the theta calculation inherently normalizes for differences in coverage. However, it may not be possible to easily extend this analysis to the case of random priming which is used in the ChemModSeq approach.

This is a great idea and we have looked into this in the past with my collaborators (Guido Sanguinetti and Alina Selega), who are expert statisticians. We tried to use this model to see if we could adopt it to our randomly primed data. It proved to be quite challenging to implement. Developing and validating the code would also take several months to complete and these analyses could be a separate paper on its own.

If that is the case, then what could be a workaround would be to normalize all reactivities to sum to 1 over the entire RNA molecule (like theta's are normalized) which should naturally remove biases associated with coverage. (Note that 2%/8% normalization does not do this.) We thus propose to perform this normalization and then repeat the analysis performed in the paper as a backup to make sure conclusions are robust. (Note that this comparison is valid for RNAs of the same length but should not be used to compare between RNAs of different lengths since length affects the normalization.)

We thank this reviewer for the helpful suggestion and this is very easy to implement. We do feel it is important to point out that the Tang et al method that we used for our analyses does correct for transcript abundance as for each sample they divide the raw drop-off counts by the average of the total drop-off counts (see page 2671 of the Tang et al 2015 Biochemistry paper, "Step 2"). We implemented the same step in our code, which is identical to what the Assmann lab have done. I pasted the code below and have annotated the regions where the coverage correction is performed (colored red).

```
def logNormalizeDropOffcounts(controldropoffs,treateddropoffs):
    """ Normalizes the drop-off counts using Tang et al 2015 approach """
    transcriptlength = float(controldropoffs.shape[0])
    assert transcriptlength == treateddropoffs.shape[0], "Coverage data and drop-off data are
    not of the same length!\n"
    logcontroldropoffs = np.log(controldropoffs + 1)      # log transform the data. 1 is added
    to also be able to include the zeros in the calculations
    logtreateddropoffs = np.log(treateddropoffs + 1)

    ### below we calculate the average drop-off counts for both the 1M7 treated
    and the control samples. We then divide all the raw counts by the average counts for
    each sample. In this way we correct for variability in coverage ###

    # calculate the average drop-offs per nucleotide

    controlaverage = sum(logcontroldropoffs)/transcriptlength
    treatedaverage = sum(logtreateddropoffs)/transcriptlength

    # normalize the counts by dividing them to the mean
    reactivitiescontrol = logcontroldropoffs/controlaverage
    reactivitiestreated = logtreateddropoffs/treatedaverage

    ###

    # calculate reactivities by subtracting modified from unmodified data
    rawreactivities = reactivitiestreated - reactivitiescontrol

    # then return the raw reactivities
    return rawreactivities
```

So assuming we interpreted their code correctly, our data is already normalized for transcript abundance. The 0-1 normalization that Reviewer 5 suggests is a great idea, however, one problem we have is that we are comparing RNA transcripts that are of different length. The 35S is almost twice the length of the 27SB transcript. Therefore the suggested normalization method does not seem applicable to our data. Secondly,

considering that our code already controls for total coverage, would the 0-1 normalization then still be necessary?

4. However, in reviewing the code used to calculate SHAPE reactivities we realized that several ad-hoc data analysis steps were performed which may need to be adjusted before trying this pseudo-theta normalization approach. Specifically, from analyzing the code it seems the following steps are performed:
 - a. Remove mapped data points with low coverage and low drop-off.
 - b. Then to deal with replicates, sum all the reads in each channel, or average (depending on options used.) The text seems to indicate that summation was used.
 - c. Then calculate reactivities according to $\text{reactivity} = \log(x+1)/\text{avg}(x) - \log(y+1)/\text{avg}(y)$ where X is what was calculated in step b. for the positive channel and Y for negative channel.
 - d. Then remove anything above $1.5 \times \text{IQR}$ before dividing everything by the NEW 10th percentile for only positive reactivities. This ends up being a not quite straightforward mix of box-plot and 2%/8% percentile that is hard to justify.
 - e. Then optionally add a 2%/8% normalization on previous step's result, but only on positive values.

It appears that step d. was not reported in the methods section of this paper and data should be re-analyzed WITHOUT steps d and a prior to doing the pseudo-theta normalization.

We agree that step 'd' is redundant and removed this it from our script. This change resulted slightly higher SHAPE reactivities overall but the SHAPE reactivity pattern (and therefore the overall conclusions) did not change. We have reanalyzed all the data with the updated code and the results are presented in the revised manuscript. The new SHAPE reactivity values are presented in Supplementary Table 2.

5. Additionally, we agree with the previous reviewer that zeros should be taken into account in the analysis.

I assume that Reviewer 5 is referring to the step where we calculated the quartiles? If we were to include the zeros when calculating the quartiles, would samples that have a higher number of zeros not have more regions called modified as the threshold is lower? In any case, the quartile classification has now been removed from the paper so this is no longer relevant.

Or is this reviewer referring to calculation of SHAPE reactivities? If so, then we can say that we are including positions with no drop-offs in our SHAPE reactivity calculations according to Tang et al. 2015. We have highlighted the regions in red below where we add 1 to all drop-off counts so that \log_2 ratios for each

position can be calculated:

```
def logNormalizeDropOffcounts(controldropoffs,treateddropoffs):
    """ Normalizes the drop-off counts using Tang et al 2015 approach """
    transcriptlength = float(controldropoffs.shape[0])
    assert transcriptlength == treateddropoffs.shape[0], "Coverage data and drop-off data are
    not of the same length!\n"

    ### log transform the data. 1 is added to also include positions that have no drop-off
    counts in the calculations.
    logcontroldropoffs = np.log(controldropoffs + 1)
    logtreateddropoffs = np.log(treateddropoffs + 1)
    ###

    ### below we calculate the average drop-off counts for both the 1M7 treated
    and the control samples. We then divide all the raw counts by the average counts for
    each sample. In this way we correct for variability in coverage ###

    # calculate the average drop-offs per nucleotide

    controlaverage = sum(logcontroldropoffs)/transcriptlength
    treatedaverage = sum(logtreateddropoffs)/transcriptlength

    # normalize the counts by dividing them to the mean
    reactivitiescontrol = logcontroldropoffs/controlaverage
    reactivitiesreated = logtreateddropoffs/treatedaverage

    ###

    # calculate reactivities by subtracting modified from unmodified data
    rawreactivities = reactivitiesreated - reactivitiescontrol

    # then return the raw reactivities
    return rawreactivities
```

6. 2.) The Weeks lab has already published a rigorous method for detecting protein binding using differential SHAPE reactivities (Smola et al. Biochemistry 2015 doi:10.1021/acs.biochem.5b00977) - see Figure 1. This analysis takes into account variability between data sets and uses a Z-score cutoff in the comparison between conditions to call positions that have different SHAPE reactivities due to protein binding. The authors should apply this method to their datasets to verify their conclusions. Note that it may be necessary to perform the normalization from suggestion 1 before performing this analysis and this should be done on reactivities that have not been classified (i.e. continuous scale reactivities).

This is an excellent suggestion. We have used their code to calculate Δ SHAPE reactivities as suggested and now present the data in the revised manuscript. We made a number of changes to the figures to accommodate the new data. The classification approach we initially described in Figure 2 is now redundant has been removed. In place of this, we now show scatter plots where we compare the results from the different probing experiments. One of the main conclusions was that most of the changes take place during the conversion of 27SA₂ to 27SB. The scatter plots clearly support this model and this is why we felt it was important to add it.

A potential problem with the Δ SHAPE algorithm is that it applies a nucleotide sliding window for its analyses and will only consider a difference in SHAPE reactivity significant several nucleotides within that window meet certain

statistical criteria (Z-factor > 0 and standard score ≥ 1) (see Figure 1 in the Smola et al 2015 paper). So in essence, whether a nucleotide is called significantly differentially modified depends on how reactive its neighbors are. This reduces noise but we also found that single nucleotide positions that showed high differential SHAPE reactivity in our previous analyses were frequently not selected in the new analyses. Therefore, a number of PKSRP sites reported in the previous version have now been removed in the revised manuscript. However, a number of inter-domain interactions formed by r-proteins we previously described (L17, L4, L34, L42 and L2) were again identified, so we decided to only highlight these in the revised Figure 4 and the main text. Unfortunately, several nucleotide positions discussed in Figure 7 were not selected by the Δ SHAPE algorithm. Because this Figure does not present key findings, we decided to remove it.

Importantly, the re-analyzed data also strongly support our main conclusions:

1. The majority of the pre-rRNA structural changes in the nucleolus take place during the conversion of 27SA₂ to 27SB.
 2. 5.8S undergoes a significant reduction in nucleotide flexibility during the conversion of 27SA₂ to 27SB (see new Figure 3b), supporting our model that this rRNA region undergoes a major conformational change at this stage.
 3. The data also support our previous findings that ITS2 is generally highly structured and does not undergo any (major) structural changes.
 4. The data identified a number of long-range r-protein-rRNA interactions that are formed during specific stages of 60S assembly, supporting the hypothesis that r-proteins gradually more stably bind to the ribosome.
7. 3.) If the authors insist on performing their classification scheme before making the comparison, then we suggest using the previously published method from SeqFold (Ouyang et al. Genome Research 2013 doi:10.1101/gr.138545.112) which performs a similar classification from reactivities to binary values (their 'structure preference profile') using more rigorous statistics than quartile classification. At least this analysis would confirm that the quartile classification is not confounding data interpretation.

We now use the Δ SHAPE code from the Weeks lab to calculate differences between particles, so a separate classification is not necessary. The Δ SHAPE data described in the current version of the manuscript enabled us to identify regions that show changes in nucleotide flexibility and overlap with r-protein binding sites (see revised Fig. 3c). These analyses also confirmed our general conclusion that folding of many domains can occur simultaneously.

General Comments:

8. In general, the impression from reading the methods section was that the authors were avoiding using the richness of the continuum reactivity scale and it was not

clear why. Hopefully these suggestions will put their analysis on a firmer footing.

We thank this reviewer for providing very helpful suggestions. Hopefully this reviewer agrees that the data analysis and methods have been improved.

9. In addition, the data analysis method section was very confusing with mentions of pipelines used in the original ChemModSeq paper AND the new Nature Methods paper but the code doesn't seem to follow the new Nature Methods paper analysis as indicated. Rather, the code appears to follow the ChemModSeq analysis despite the authors saying that they did NOT use it in this present work. This needs to be clarified.

We apologize for the confusion. When we refer to the ChemModSeq pipeline, we refer to the pipeline required to process the raw fastq files into files containing drop-off and read-counts for each nucleotide (.sgr files). We then use these output files to calculate SHAPE reactivities or to calculate the probability that a nucleotide is modified (BUM_HMM). In the methods section of the manuscript we now explain in detail what data processing steps our ChemModSeq pipeline performs and we clearly state that SHAPE reactivities were calculated using a separate script.

10. Furthermore, the authors state in the new Nature Methods paper that:

"Our statistical pipeline addresses a number of important problems in the analysis of high-throughput RNA secondary structure probing data. First, it explicitly models the biological variability of the data, providing a statistical basis for determining the significance of the observed signal. As such, it removes the need to set arbitrary thresholds and perform extensive postprocessing of the analysis results, yielding a clean and statistically interpretable pipeline." The last sentence begs the question as to why the authors are doing the various processing and thresholding steps reported here. Perhaps they should rely on the method described in their Nature Methods paper to also repeat these analyses to make sure their conclusions are robust.

The BUM_HMM output provides posterior probabilities that indicate the likelihood that a nucleotide is modified. The output (values between 0 and 1) are meant to be used as constraints to improve the secondary structure predictions by software such as RNA-fold and RNAstructure. However, the output cannot (yet) be used to compare different particles as we have done here. In order to do a binary classification of the BUM_HMM data we would still have to set a threshold for a minimum posterior probability. In addition, we cannot use the BUM_HMM values to calculate differences in reactivity as it does not provide a reactivity value, and subtracting posterior probabilities does not make any statistical sense. So as it stands, the method cannot be used to compare different particles. We are currently working hard to adapt the BUM_HMM code to make this possible by not only asking it to compare modified to unmodified samples, but also to

consider different samples. With the new BUM_HMM code we aim to generate posterior probabilities indicating the probability that a nucleotide between samples has a different reactivity. However, this is not trivial to implement and would go beyond the scope of the current manuscript. This is why we decided to use an analysis method that was previously published in two highly regarded journals (Ding et al. *Nature* 2014; Tang et al. *Bioinformatics* 2015).

REVIEWERS' COMMENTS:

Reviewer #5 (Remarks to the Author):

In this revised manuscript the authors have re-analyzed their data according to several suggestions made by this reviewer. In particular, the authors found the SHAPE analysis for detecting differences in SHAPE reactivities between multiple experimental conditions developed by the Weeks lab was compatible with their experimental approach. Importantly, the authors show that this new data analysis scheme produced conclusions that were in line with their previous approaches, giving extra evidence in support of their biological conclusions.

We thank the authors for spending the extra time to review this reviewer's suggestions and to implement them – their commitment to performing additional data analysis to support their claims is very much appreciated. In addition, we thank them for the clarity in their response including specific code snippets to demonstrate their approaches.

We now feel that this manuscript is acceptable for publication.